# DENOISING WITH A JOINT-EMBEDDING PREDICTIVE ARCHITECTURE

**Dengsheng Chen**[1]    **Jie Hu**[1,2]    **Xiaoming Wei**[1]    **Enhua Wu**[2*]
[1]Meituan    [2]Key Laboratory of System Software (Chinese Academy of Sciences) and
State Key Laboratory of Computer Science, Institute of Software, Chinese Academy of Sciences
{chendengsheng, weixiaoming}@meituan.com, {hujie, weh}@ios.ac.cn

## ABSTRACT

Joint-embedding predictive architectures (JEPAs) have shown substantial promise in self-supervised representation learning, yet their application in generative modeling remains underexplored. Conversely, diffusion models have demonstrated significant efficacy in modeling arbitrary probability distributions. In this paper, we introduce Denoising with a Joint-Embedding Predictive Architecture (D-JEPA), pioneering the integration of JEPA within generative modeling. By recognizing JEPA as a form of masked image modeling, we reinterpret it as a generalized next-token prediction strategy, facilitating data generation in an autoregressive manner. Furthermore, we incorporate diffusion loss to model the per-token probability distribution, enabling data generation in a continuous space. We also adapt flow matching loss as an alternative to diffusion loss, thereby enhancing the flexibility of D-JEPA. Empirically, with increased GFLOPs, D-JEPA consistently achieves lower FID scores with fewer training epochs, indicating its good scalability. Our base, large, and huge models outperform all previous generative models across all scales on ImageNet conditional generation benchmarks. Beyond image generation, D-JEPA is well-suited for other continuous data modeling, including video and audio.

Project page: `https://d-jepa.github.io/`.

## 1 INTRODUCTION

The extraordinary success of generative language models (Radford et al., 2018; 2019; Brown et al., 2020), which excel in next-token prediction, has spurred efforts to adapt autoregressive models for other domains, notably image generation (Kilian et al., 2024; Zeng et al., 2021; Li et al., 2024; Esser et al., 2021; Yu et al., 2023b). Typically, this involves discretizing visual data into token sequences (Van Den Oord et al., 2017), a process that can sometimes degrade image quality. Despite this, the technique has gained substantial interest due to its powerful capabilities (Razavi et al., 2019; Peng et al., 2021; Yan et al., 2021).

Recently, diffusion models (Ho et al., 2020; Song et al., 2020; Nichol & Dhariwal, 2021) have emerged as a dominant force in generative modeling, surpassing generative adversarial networks (GANs) (Goodfellow et al., 2014; Wang et al., 2017; Goodfellow et al., 2020) with their superior performance in producing high-fidelity images. Concurrently, significant advancements have been made in self-supervised representation learning, particularly through masked image modeling (Xie et al., 2022; Baevski et al., 2022; Assran et al., 2023; He et al., 2022a) and invariance-based techniques (Assran et al., 2022; Caron et al., 2021; Chen & He, 2021; Grill et al., 2020; Zbontar et al., 2021). However, generative modeling has yet to directly benefit from the advancements in representation learning (Li et al., 2023).

Recent efforts have sought to integrate diffusion models within an autoregressive framework, such as MAR (Li et al., 2024) and Transfusion (Zhou et al., 2024). These models, however, primarily focus on enhancing generative model performance from the perspective of generation alone, often overlooking the potential advancements that can be brought by representation learning. Conversely,

---

*This work is supported in part by NSFC Grants (62332015).

within the realm of representation learning, researchers tend to emphasize model performance on downstream tasks (Assran et al., 2023), neglecting the potential for these models to excel in generative modeling. To overcome this challenge, we propose a novel framework, termed "Denoising with a Joint-Embedding Predictive Architecture" (D-JEPA), that effectively leverages next-token prediction from a representation learning perspective to generative modeling. Specifically, we adopt the joint-embedding predictive architecture as the core of our design, enabling us to reframe masked image modeling methods into a generalized next-token prediction strategy, thus facilitating autoregressive data generation. Additionally, we incorporate a diffusion loss (or flow matching loss) to model the per-token probability distribution, allowing for data generation in a continuous space.

D-JEPA consists of three identical visual transformer backbones (Dosovitskiy et al., 2020): a context encoder $\phi$, a target encoder $\bar{\phi}$, and a feature predictor $\gamma$. It employs two loss functions: diffusion loss ($\mathcal{L}_d$) and prediction loss ($\mathcal{L}_p$), both involving a multi-layer perceptron (MLP) network. The denoising MLP, used in diffusion loss, is applied to each masked token $x_i$, modeling the conditional probability distribution $p(x_i \mid z_i)$ and evaluating the predicted latent variable $z_i$ generated by the feature predictor $\gamma$. The prediction loss ($\mathcal{L}_p$), using a smooth loss function $l_1$, facilitates the latent variables $z_i$ to capture high-level semantic information.

Empirically, D-JEPA exhibits a consistent decrease in FID scores as GFLOPs increase, achieving these results with fewer training epochs, indicating its good scalability. Our base, large, and huge variants outperform all previously generative models across different scales on ImageNet conditional generation benchmarks. In our comprehensive experiments, we further advanced the field of generative models by substituting the diffusion loss with flow matching loss (Lipman et al., 2022). This substitution enables D-JEPA to excel in image generation and modeling other types of continuous data, including video and audio. Moreover, D-JEPA exhibits remarkable flexibility, extending seamlessly to text-conditioned and multi-modal generations.

Our contributions are threefold. *Firstly*, we successfully integrate advancements in representation learning into generative modeling, achieving state-of-the-art performance on the ImageNet conditional generation benchmarks. *Secondly*, we validate the effectiveness of the generalized next-token prediction approach in generative modeling, demonstrating superior scalability, which establishes a robust foundation for developing more powerful text-conditioned models. *Lastly*, D-JEPA exhibits impressive extensibility, directly applicable to generate images, videos, audio, and more, providing theoretical support for constructing unified multi-modal generation models for continuous data.

## 2 BACKGROUND

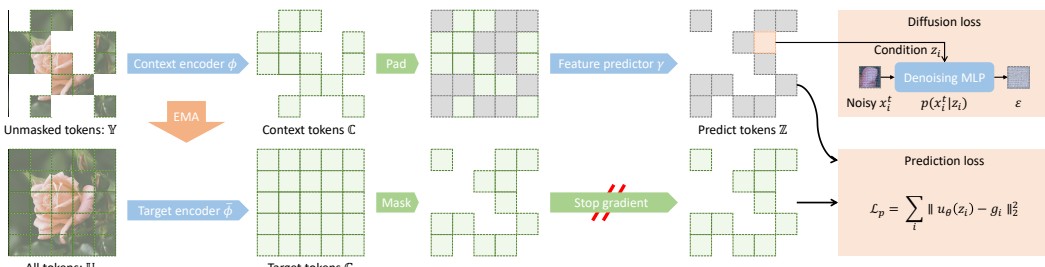

Figure 1: Data flow during D-JEPA training. Initially, the training data is divided into non-overlapping semantic tokens, which can be either in the raw space or in the latent space obtained after VAE encoding. A random subset of these input tokens is then masked. The feature predictor $\gamma$ is employed to predict features for these masked tokens, utilizing the unmasked tokens as contextual information. Each masked token is concurrently subjected to a diffusion loss (or flow matching loss) to learn the distribution of each token $p(x_i|z_i)$, independently. Additionally, a prediction loss is applied, compelling each masked token to regress towards the target tokens $g_i$.

D-JEPA is grounded in the principles of joint-embedding predictive architectures (JEPAs) and diffusion models. We provide an overview of these methodologies alongside a generalized next-token prediction strategy to facilitate an understanding of our approach.

**Joint-embedding predictive architectures (JEPAs).** Self-supervised learning has transformed representation learning by enabling systems to uncover autonomously and model relationships within their inputs (LeCun et al., 2006). Among these methods, joint-embedding predictive architectures (JEPAs)(LeCun, 2022; Bardes et al., 2023a;b; Guetschel et al., 2024) have gained prominence. JEPAs are designed to predict the embeddings of a target signal using a context signal processed through a predictor network, which is conditioned on an additional (potentially latent) variable, such as class labels.

A significant challenge in deploying JEPAs is representation collapse, which occurs when the energy landscape flattens, causing the encoder to produce uniform outputs regardless of the inputs. This uniformity negates the advantages of learned representations (Assran et al., 2023). Various strategies have been proposed to address this issue. Contrastive losses (He et al., 2019; Chen & He, 2021) work by explicitly separating the embeddings of negative samples. Non-contrastive losses (Zbontar et al., 2021; Bardes et al., 2021) aims to reduce redundancies between embeddings, thereby improving robustness. Clustering-based methods (Caron et al., 2021; Assran et al., 2021; 2022) increase the entropy of the aggregate embedding to maintain diversity. Additionally, heuristic approaches (Chen & He, 2021; Grill et al., 2020; Baevski et al., 2022; Assran et al., 2023) often utilize asymmetric architectural designs between the context encoder $\phi$ and the target encoder $\bar{\phi}$ to prevent collapse.

However, the issue of representation collapse is effectively mitigated in D-JEPA due to the introduction of an additional diffusion loss for the predicted embeddings. The diffusion loss does not reside in the same energy landscape as JEPAs, thus effectively avoiding model collapse. Consequently, similar to the approach in LeCun (2022), we can employ two identical networks as the context and target encoders. The parameters of the target encoder are efficiently updated through an exponential moving average method, significantly enhancing training efficiency and reducing the number of parameters that need to be learned. Please refer to Sec. 3.2 for more details.

**Diffusion models.** Denoising diffusion models provide a robust framework for modeling arbitrary distributions (Croitoru et al., 2023). Given a ground truth sample $x$ conditioned on a variable $z$, the objective is to model the conditional probability distribution $p(x|z)$. The sample $x$ could represent various domains, such as images in pixel space, features in latent space, or control signals in robotic systems (Chi et al., 2023; Team et al., 2024). In our case, $x$ represents a token, while $z$ contains information to aid in reconstructing $x$; here, $z$ corresponds to the predicted embeddings by feature predictor $\gamma$.

Following the formulation presented in Dhariwal & Nichol (2021), we define the loss function for modeling the underlying probability distribution $p(x|z)$ using a denoising criterion:

$$\mathcal{L}(z, x) = \mathbb{E}_{\varepsilon, t}\left[\left\|\varepsilon - \varepsilon_\theta(x^t|t, z)\right\|^2\right],\tag{1}$$

where $\varepsilon$ represents a noise sample from the normal distribution $\mathcal{N}(\mathbf{0}, \mathbf{I})$. The noise-corrupted sample $x^t$ is expressed as $x^t = \sqrt{\bar{\alpha}_t}x + \sqrt{1 - \bar{\alpha}_t}\varepsilon$, with $\bar{\alpha}_t$ defining a noise schedule (Ho et al., 2020; Nichol & Dhariwal, 2021) and $t$ indicating a specific time step within this schedule. The function $\varepsilon_\theta(x^t|t, z)$ denotes the network processing $x^t$, conditioned on both $t$ and $z$. As discussed in Song & Ermon (2019) and Song et al. (2021), Eq. 1 acts as a form of score matching, associated with a loss function regarding the score function of $p(x|z)$, represented as $\nabla \log_x p(x|z)$.

Unlike typical applications of diffusion models that represent the joint distribution of all pixels or tokens, our approach leverages the diffusion model to represent the distribution of individual tokens following Li et al. (2024), *i.e.*, $p(x_i|z_i)$. This allows us to simultaneously apply both diffusion loss and prediction loss to the tokens. Consequently, the noise estimator $\varepsilon_\theta$, parameterized by $\theta$, can be efficiently modeled using a small MLP network in our case. This significantly enhances the efficiency of the denoising process in the continuous space. Further details are provided in Sec. 3.3.

**Generalized next-token prediction.** Next-token prediction, or autoregressive modeling, typically predicts the next token based on preceding tokens, establishing a sequential dependency (Radford et al., 2018). Generalized next-token prediction techniques, such as MaskGIT (Chang et al., 2022) and MAGE (Li et al., 2023), integrate an MAE-style (He et al., 2022a) encoder and decoder with positional embeddings to streamline the prediction process. This approach facilitates interaction among known tokens and accelerates inference by generating multiple tokens per step, as demon-

strated in the next set-of-token prediction strategies like MAR (Li et al., 2024), even without key-value caching (Shazeer, 2019).

Our D-JEPA model extends these innovations by integrating the masked image modeling approach as a generalized next-token prediction strategy to facilitate an autoregressive generation manner. We can flexibly adjust the number of tokens predicted in each step by controlling the auto-regressive steps. Contrary to the intuition formed by existing methods (Li et al., 2024; 2023), the best performance under the D-JEPA architecture is not achieved when the auto-regressive steps are maximized, *i.e.*, when the generalized next-token prediction predicts only one token at a time. Moreover, as the scale of the model parameters increases, the number of auto-regressive steps required to achieve the best sampling results decreases accordingly. This fully demonstrates that converting the masked image modeling method into a next-token prediction strategy is not only feasible but also highly effective. Please refer to the experiment part in Sec. 4 for more analysis.

## 3 METHODOLOGY

In this section, we will first introduce how to convert masked image modeling into the next-token prediction. Following this, we will explain how representation learning and diffusion learning co-exist within D-JEPA. Finally, we will describe how to generate samples using the generalized next-token prediction approach. The overall architecture of D-JEPA is illustrated in Fig. 1.

For clarity, we denote a sequence of tokens using blackboard bold letters, such as $\mathbb{X}$, with individual tokens represented by lowercase letters with subscripts (*e.g.*, $x_i$ as a token in $\mathbb{X}$). The notation $|\mathbb{X}|$ indicates the number of tokens in the sequence.

### 3.1 TOKENIZATION AND MASKING

In masked image modeling, certain image regions are randomly masked during training, prompting the feature predictor $\gamma$ to predict the content of these masked areas based on the context (unmasked) regions. In representation learning, masking is typically applied directly to raw pixels (He et al., 2022a; Assran et al., 2023), and it has been observed that masking contiguous regions significantly enhances the representation learning capability. This is crucial, as it implies that when we patchify an image into tokens, we can randomly mask tokens without degrading performance since each token inherently represents a continuous region. Based on this, we can patchify images into non-overlapping semantic tokens $\mathbb{U}$ based on a specific patch size $p$.

Although images are directly segmented into patches on raw pixels, which is also the standard practice in visual transformers (Dosovitskiy et al., 2020), recent generative models (Li et al., 2024; 2023; Rombach et al., 2022) tend to first encode images into the latent space using a VAE before performing patchification, arguing that this significantly reduces the training cost of generative models and accelerates the sampling speed. However, we found that this approach noticeably degrades D-JEPA's performance in representation tasks. Since the primary goal of this work is to ensure that D-JEPA functions as an outstanding generative model, we adopted the latent space modeling strategy initially proposed by Rombach et al. (2022). (It is important to note that *training in latent space is not a necessity for D-JEPA*; it can be trained in raw space and still achieve excellent results. The experiments conducted in raw pixel space are detailed in Appendix E.2.)

Subsequently, we perform random masking on these tokens. Masking ratios $r_{mask}$ are sampled from a truncated normal distribution with a mean of 1.0, a standard deviation of 0.25, and a lower bound of 0.7. Consequently, more than 70% of the tokens are randomly masked. This results in a sequence of randomly masked tokens of length $\lceil r_{mask} \cdot |\mathbb{U}| \rceil$, and a sequence of remaining unmasked tokens, $\mathbb{Y}$. Masking a substantial portion of tokens reduces both pre-training time and memory usage while improving performance, consistent with findings in previous works(He et al., 2022a; Li et al., 2023; Assran et al., 2023).

### 3.2 REPRESENTATION LEARNING WITH JEPAS

Earlier masked image modeling methods, such as MAE (He et al., 2022a), directly use a decoder network to predict the missing raw pixels. This approach gives these models some outpainting capabilities (i.e., completing missing parts of an image), but they still cannot directly generate high-

quality images. Subsequent works, such as DiffMAE (Wei et al., 2023), attempted to leverage diffusion models within MAE (He et al., 2022a) to enhance the image generation capabilities of these models, but the results were limited. Moreover, this direct raw pixel prediction approach severely restricts the model's representation ability, especially compared to models like CLIP (Radford et al., 2021), which use contrastive learning on massive datasets.

Meanwhile, a new architecture called JEPA has gained widespread attention and was successfully applied to representation learning on ImageNet by Assran et al. (2023), achieving excellent results. JEPA does not directly predict the missing raw pixels; instead, it uses a separate feature predictor $\gamma$ to predict the target embedding $z_i$ corresponding to the missing parts. The target embedding $g_i$, generated by the target encoder $\bar{\phi}$ by feeding all tokens ($\mathbb{U}$), is used as the ground truth for $z_i$.

The prediction loss objective for unsupervised representation learning on each masked token is defined as:

$$\mathcal{L}_p = \mathbb{E}_{z_i=\psi(c_i),g_i} \left[ \mathcal{D}(u_\theta(z_i), g_i) \right], \tag{2}$$

where $\mathcal{D}(u_\theta(z_i), g_i)$ is a distance measure function, and $u_\theta(z_i)$ is the projection feature of $z_i$ using a two-layer MLP $u_\theta$. This loss is optimized exclusively for masked tokens, as optimizing for all tokens has been shown to diminish both generative and representational performance, as noted in He et al. (2022a); Li et al. (2023; 2024). In fact, the choice of distance measure function $\mathcal{D}$ in Eq. 2 is relatively flexible. In Assran et al. (2023), the MSE function was chosen, while Bardes et al. (2023a) used the $l_1$ function. Here, we use the smoothed $l_1$ function, following Baevski et al. (2022), which we found to be more stable.

Since JEPA predicts feature embeddings, they cannot be directly used for generation tasks or even outpainting. Experiments show that token embeddings trained with JEPA can only reconstruct image content somewhat related to the surrounding context and are almost incapable of generating high-quality images (Bardes et al., 2023a). This further diminishes the consideration of JEPA architecture as a generative model. Here, we surprisingly find that as long as the JEPA training process also includes generative task training, excellent generative results can be achieved, which will be elaborated on in the next section.

We use three identical visual transformers as the context encoder $\phi$, target encoder $\bar{\phi}$, and feature predictor $\gamma$. The parameters of $\phi$ and $\gamma$ are randomly initialized and then updated with a gradient descent method. The parameters of the target encoder are initialized identically to the context encoder and then updated via an exponentially moving average of the context encoder parameters:

$$\bar{\phi} \leftarrow \alpha\bar{\phi} + (1-\alpha)\phi, \tag{3}$$

where $\alpha$ controls the update frequency. In Bardes et al. (2023a) and Assran et al. (2023), a linear warmup is applied to gradually increase $\alpha$ from a smaller value to a larger one to effectively control the update speed of the target encoder, thereby preventing the model collapse. Thanks to the constraint of diffusion loss, for D-JEPA, we simply use a constant $\alpha = 0.9999$, which is sufficient.

In Eq. 2, $g_i = \text{sg}(\bar{\phi}(\mathbb{U}))$, where $\text{sg}(\cdot)$ denotes a stop-gradient operation, which does not backpropagate through its argument. This ensures that the parameters of the target encoder $\bar{\phi}$ are updated only through exponentially moving average , *i.e.*, Eq. 3, a strategy that has been used to prevent collapse in image pre-training (Grill et al., 2020), and studied empirically (Xie et al., 2022) and theoretically (Tian et al., 2021) [1].

### 3.3 DIFFUSION LEARNING WITH A DENOISING MLP

Our method diverges from traditional approaches by focusing on learning the conditional distribution $p(x_i|z_i)$ for each token, given $z_i$, rather than capturing the distribution of entire images. This approach eliminates the need to denoise the entire JEPA model, as some prior works require Hatamizadeh et al. (2023); Peebles & Xie (2023); Wei et al. (2023), allowing us to utilize a significantly smaller network to model each token individually.

Hence, the denoising loss objective can be formulated by slightly modifying Eq. 1 to model the token distribution as:

$$\mathcal{L}_d = \mathbb{E}_{\varepsilon,t} \left[ \left\| \varepsilon - \varepsilon_\theta(x_i^t|t, z_i) \right\|^2 \right]. \tag{4}$$

---

[1] A theoretical motivation for the effectiveness of this collapse prevention strategy is proposed in Appendix E.1.

For denoising, we employ a compact multi-layer perceptron (MLP) comprising a few residual blocks (He et al., 2016) as $\varepsilon_\theta$. Each block sequentially applies LayerNorm (LN) (Ba et al., 2016), a linear layer, the SiLU activation function (Elfwing et al., 2018), and another linear layer, integrated with a residual connection, following Li et al. (2024). $z_i$ is added to the time embedding of the noise schedule time-step $t$, then conditions the MLP in the LN layers via AdaLN (Peebles & Xie, 2023). During training, we independently sample $t$ four times for each token to maximize the utilization of the diffusion loss without recomputing $z_i$.

At inference time, it is required to draw samples from the distribution $p(x_i|z_i)$. Sampling is done via a reverse diffusion procedure (Ho et al., 2020):

$$x_{t-1}^i = \frac{1}{\sqrt{\alpha_t}}\left(x_t^i - \frac{1-\alpha_t}{\sqrt{1-\bar{\alpha}_t}}\varepsilon_\theta(x_t^i|t, z_i)\right) + \sigma_t\delta,$$

where $\delta$ is sampled from the Gaussian distribution $\mathcal{N}(\mathbf{0}, \mathbf{I})$ and $\sigma_t$ is the noise level at time step $t$. Starting with $x_T^i \sim \mathcal{N}(\mathbf{0}, \mathbf{I})$, this procedure produces a sample $x_0^i$ such that $x_0^i \sim p(x_i|z_i)$.

We adopt the temperature sampling method presented in Dhariwal & Nichol (2021); Li et al. (2024). Conceptually, with a temperature $\tau$, one would sample from the (renormalized) probability $p(x_i|z_i)^{\frac{1}{\tau}}$, whose score function is $\frac{1}{\tau}\nabla \log p(x_i|z_i)$. Intuitively, we adjust the noise variance $\sigma_t\delta$ in the sampler by $\tau$ to control the sample diversity. Experimentally, $\tau = 0.98$ and $\tau = 0.94$ are suitable for image generation with and without classifier-free guidance, respectively.

Furthermore, we also introduce the recently popular *flow matching loss* (Lipman et al., 2022), *i.e.*, Eq. 7, as an alternative to the denoising loss (Eq. 4). Overall, flow matching achieves faster convergence and generates higher-quality visual content, and it can be more easily adapted to different scenarios. However, for scenarios where the diffusion loss can converge, fully trained diffusion loss can achieve better performance than flow-matching loss. [2]

## 3.4 Training with D-JEPA

By choosing different training objectives, D-JEPA can seamlessly transition between representation learning and diffusion learning without any changes to the network architecture. Typically, when faced with multiple training objectives, one needs to carefully balance each component in the final loss objective to ensure the model's effectivenes (Kendall et al., 2018). However, the prediction loss $\mathcal{L}_p$ and the diffusion loss $\mathcal{L}_d$ in D-JEPA do not have conflicting or mutually exclusive relationships.

The diffusion loss effectively prevents the potential representation collapse issue faced by the prediction loss, while the prediction loss can provide the diffusion model with higher-level semantic feature information $z_i$. Therefore, the overall loss function can be simply defined as:

$$\mathcal{L} = \mathcal{L}_d + \mathcal{L}_p.$$

Experiments show that this approach plays a crucial role in enabling D-JEPA to converge faster and achieve better results when scaling up the model.

## 3.5 Sampling in Next Set-of-Tokens Prediction

For the evaluation of generative models in generalized next-token prediction, we adopt an *iterative sampling* strategy analogous to those used in Chang et al. (2022); Li et al. (2023; 2024), as shown in Algo. 1. This strategy gradually reduces the masking ratio from $1.0$ to $0.0$ in accordance with a cosine schedule, typically utilizing $64$ auto-regressive steps. D-JEPA employs fully randomized orderings following Li et al. (2024) to decide the next set of tokens to predict. This design, along with the temperature $\tau$, effectively enhances the diversity of the generated samples.

When the number of auto-regressive steps $T$ equals the total tokens $N$, it means that we sample one token at each step. In this case, the *generalized next-token prediction* is equivalent to a *typical next-token prediction*. In another extreme case, $T = 1$, all tokens are sampled in one step, *i.e.*, *one-step generation* like He et al. (2022a).

---

[2]Here, we discuss flow matching loss and diffusion loss, which are not equivalent to flow matching models and diffusion models. For a detailed analysis, please refer to Appendix F.

As analyzed in Appendix C.2, for D-JEPA, adopting a more efficient next set-of-tokens prediction strategy often yields better results. Particularly, as the model scale increases, fewer auto-regressive steps are required. Additionally, as shown in Fig. 5, one-step generation fails to produce high-quality images, which is a major reason for the subpar performance of MAE (He et al., 2022a). Even with just eight steps of iterative sampling, we can generate clearly recognizable content. This fully demonstrates that D-JEPA's use of the next set of tokens prediction strategy is key to ensuring its strong generative capabilities.

---

**Algorithm 1** Generalized next token prediction with D-JEPA

---

**Require:** $T$: Number of auto-regressive steps, $N$: Total tokens to sample, $\tau$: Temperature to control noise.

1: **Initialize:** $\mathbb{X} \leftarrow \emptyset$
2: **for** $n$ in cosine-step-function$(T, N)$ **do**
3:      $\mathbb{C} \leftarrow \phi(\mathbb{X})$          ▷ Encode the sampled tokens to obtain context features
4:      $\mathbb{Z} \leftarrow \gamma(\mathbb{C})$          ▷ Predict features of unsampled tokens using the feature predictor
5:      $\{z_0, \ldots, z_n\} \sim \mathbb{Z}$          ▷ Randomly select $n$ tokens from $\mathbb{Z}$
6:      $\{x_0, \ldots, x_n\} \leftarrow \text{denoise}(\epsilon_\theta, \{z_0, \ldots, z_n\}, \tau)$    ▷ Perform denoising on the selected tokens
7:      $\mathbb{X} \leftarrow \mathbb{X} \cup \{x_0, \ldots, x_n\}$          ▷ Add the denoised tokens to $\mathbb{X}$
8: **end for**
9: **Return:** $\mathbb{X}$

---

## 4 EXPERIMENTS

To evaluate D-JEPA's generative performance, we conduct experiments on the ImageNet-1K dataset Russakovsky et al. (2015) for the task of class-conditional image generation. Please refer to Appendix B for detailed experimental settings.

### 4.1 IMAGE SYNTHETIS

| | #Params | #Epochs | FID↓ | IS↑ | Pre.↑ | Rec.↑ |
|---|---|---|---|---|---|---|
| *Base scale model (less than 300M)* | | | | | | |
| MAR-B (2024) | 208M | 800 | 3.48 | 192.4 | 0.78 | 0.58 |
| D-JEPA-B | 212M | 1400 | **3.40** | **197.1** | 0.77 | **0.61** |
| MaskGIT (2022) | 227M | 300 | 6.18 | 182.1 | **0.80** | 0.51 |
| MAGE (2023) | 230M | 1600 | 6.93 | 195.8 | - | - |
| *Large scale model (300∼700M)* | | | | | | |
| GIVT (2023) | 304M | 500 | 5.67 | - | 0.75 | 0.59 |
| MAGVIT-v2 (2023b) | 307M | 1080 | 3.65 | 200.5 | - | - |
| MAR-L (2024) | 479M | 800 | 2.60 | 221.4 | **0.79** | 0.60 |
| DiT-XL (2023) | 675M | 1400 | 9.62 | 121.5 | 0.67 | **0.67** |
| SiT-XL (2024a) | 675M | 1400 | 8.60 | - | - | - |
| MDTv2-XL (2023) | 676M | 400 | 5.06 | 155.6 | 0.72 | 0.66 |
| D-JEPA-L | 687M | 480 | **2.32** | **233.5** | **0.79** | 0.62 |
| *Huge scale model (900+M)* | | | | | | |
| MAR-H (2024) | 943M | 800 | 2.35 | 227.8 | **0.79** | **0.62** |
| D-JEPA-H | 1.4B | 320 | **2.04** | **239.3** | **0.79** | **0.62** |
| VDM++ (2024) | 2.0B | 1120 | 2.40 | 225.3 | - | - |

Table 1: System-level comparison on ImageNet $256 \times 256$ conditional generation *without* classifier-free guidance. For ease of comparison, we have omitted earlier works with FID greater than 10. A more detailed table can be found in the appendix.

**Quantitative analysis.** We present the Fréchet Inception Distance (FID) along with other metrics in Tab. 1 without classifier-free guidance. Two critical conclusions can be drawn from the results: *State-of-the-Art Performance*: D-JEPA achieves state-of-the-art FID and Inception Score (IS) across all model scales. In particular, D-JEPA-L, with only 687M parameters, surpasses all

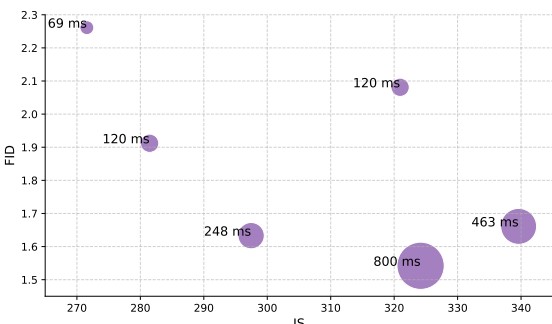

Figure 2: FID *vs*. IS under different sampling efficiencies. We can achieve FID $\approx 4.0$ within 43 milliseconds (not plotted in the figure for compactness). Additionally, the figure shows that D-JEPA can achieve FID $\approx 2.0$ within 120 milliseconds. Refer to Table 4 for more details. All times are the average inference time using a batch size of 256 on a single H800 GPU for generating $256 \times 256$ images.

previous works across all scales with FID $= 2.32, \text{IS} = 233.5$. The larger model, D-JEPA-H, further improves the benchmark set by D-JEPA-L to FID $= 2.04, \text{IS} = 239.3$. *Efficiency in Training*: As the model scale increases, the required number of training epochs for D-JEPA decreases significantly. Both D-JEPA-L and D-JEPA-H achieve state-of-the-art performance with substantially fewer training epochs compared to other models. For instance, D-JEPA-L, with 687M parameters and 480 training epochs, outperforms the previous state-of-the-art model, MAR-H, which has 943M parameters and requires 800 training epochs. It is worth noting that D-JEPA-B achieved similar performance to MAR-B (FID=3.50 vs. 3.48) with only 600 training epochs compared to MAR-B's 800 epochs. For a more detailed analysis, please refer to the ablation study on Appendix C.3.

It is well known that selecting an appropriate cfg scale (Ho & Salimans, 2021) can significantly enhance the quality of generated images. For D-JEPA, in addition to adjusting the cfg scale, it is also necessary to choose an appropriate temperature $\tau$ to ensure optimal generation quality. We employ a coarse-to-fine grid search to determine the optimal sampling parameters, as detailed in the Appendix C.2. The optimal sampling configurations are listed on Tab. 5.

We list additional results in Tab. 2 with classifier-free guidance. To better showcase D-JEPA's performance, we provide two sets of sampling performance metrics for each model scale, representing the optimal FID and settings more inclined towards IS. The table reveals that D-JEPA achieves superior results in this scenario as well. Specifically, D-JEPA-L and D-JEPA-H reach FID scores of 1.58 and 1.54, respectively, which are very close to the upper limit of the used VAE (FID = 1.199). Although MAR-H achieves a comparable FID (1.55), its IS (303.7) is significantly lower than that of D-JEPA-L (327.2) and D-JEPA-H (341.0), and it requires more training epochs.

**Qualitative analysis.** In Fig. 3, we present images generated using the next set-of-tokens prediction. Even when trained on ImageNet1k, D-JEPA is capable of producing highly realistic images with rich and clear local details. Notably, the figure also demonstrates that D-JEPA-H can generate high-quality portraits (the second picture in the second row), a capability not seen in previous generative models. Additionally, as shown in Fig. 5, we observe the images generated at different auto-regressive steps. Even with just 8 steps, D-JEPA demonstrates significant generative capabilities, and at 64 steps, it nearly matches the image quality achieved at 256 steps. Fewer auto-regressive steps imply faster image generation, which is crucial for practical applications. In Fig. 2, we use a bubble chart to illustrate D-JEPA's performance under different computational costs. Remarkably, we can sample high-quality images with FID $\approx 4.0$ in just 43 milliseconds, which significantly outperforms previous diffusion models and auto-regressive models.

**Scaling law of D-JEPA.** In summary, as the parameter scale increases, D-JEPA achieves better performance with fewer training epochs, demonstrating excellent scalability. Although similar conclusions have been observed in Peebles & Xie (2023) and Li et al. (2024), for the former, the required training epochs increase with model size, and for the latter, the required training epochs do not decrease with model size.

Moreover, with increasing model size, we are surprised to find that the number of auto-regressive steps required for D-JEPA to achieve optimal sampling results also significantly decreases[3]. Specif-

---

[3]Refer to Appendix C.2 for more details.

|  | #params | FID↓ | IS↑ | Pre.↑ | Rec.↑ |
|---|---|---|---|---|---|
| *Base scale model (less than 300M)* | | | | | |
| MAR-B (2024) | 208M | 2.31 | 281.7 | **0.82** | 0.57 |
| D-JEPA-B(cfg=3.1) | 212M | **1.87** | 282.5 | 0.80 | **0.61** |
| D-JEPA-B(cfg=4.1) | 212M | 2.08 | **320.9** | **0.82** | 0.59 |
| *Large scale model (300∼700M)* | | | | | |
| GIVT (2023) | 304M | 3.35 | - | 0.84 | 0.53 |
| MAGVIT-v2 (2023b) | 307M | 1.78 | 319.4 | - | - |
| VAR-d16 (2024) | 310M | 3.30 | 274.4 | 0.84 | 0.51 |
| LDM-4 (2022) | 400M | 3.60 | 247.7 | **0.87** | 0.48 |
| MAR-L (2024) | 479M | 1.78 | 296.0 | 0.81 | 0.60 |
| U-ViT-H (2022) | 501M | 2.29 | 263.9 | 0.82 | 0.57 |
| ADM (2021) | 554M | 4.59 | 186.7 | 0.82 | 0.52 |
| Flag-DiT (2024b) | 600M | 2.40 | 243.4 | 0.81 | 0.58 |
| Next-DiT (2024a) | 600M | 2.36 | 250.7 | 0.82 | 0.59 |
| VAR-d20 (2024) | 600M | 2.57 | 302.6 | 0.83 | 0.56 |
| DiT-XL (2023) | 675M | 2.27 | 278.2 | 0.83 | 0.57 |
| SiT-XL (2024a) | 675M | 2.06 | 270.2 | 0.82 | 0.59 |
| MDTv2-XL (2023) | 676M | **1.58** | 314.7 | 0.79 | **0.65** |
| D-JEPA-L(cfg=3.0) | 687M | **1.58** | 303.1 | 0.80 | 0.61 |
| D-JEPA-L(cfg=3.9) | 687M | 1.65 | **327.2** | 0.81 | 0.61 |
| *Huge scale model (900+M)* | | | | | |
| MAR-H (2024) | 943M | 1.55 | 303.7 | 0.81 | **0.62** |
| VAR-d24 (2024) | 1.0B | 2.09 | 312.9 | 0.82 | 0.59 |
| D-JEPA-H(cfg=3.9) | 1.4B | **1.54** | 324.2 | 0.81 | **0.62** |
| D-JEPA-H(cfg=4.3) | 1.4B | 1.68 | **341.0** | **0.82** | 0.61 |
| VDM++ (2024) | 2.0B | 2.12 | 267.7 | - | - |

Table 2: System-level comparison on ImageNet $256 \times 256$ conditional generation.

ically, for D-JEPA-H, we achieve the best sampling results with just 64 steps. This property ensures performance for constructing larger models and generating ultra-high-definition images and videos such as Sora (OpenAI, 2024) in the future.

## 4.2 COMPREHENSIVE EXPERIMENTS

In Appendix D.4, we further demonstrate the excellent inpainting and outpainting capabilities of the D-JEPA model. Beyond diffusion loss, we also designed a flow matching loss to construct a more powerful D-JEPA (Appendix F). Experiments show that flow matching loss performs better on many tasks. Consequently, we extended our experiments based on flow matching loss to include text-to-image/audio generation and class-conditional video generation. These extended experiments fully demonstrate that D-JEPA is capable of generating various types of continuous data, not limited to images. Furthermore, we attempted to construct a multimodal generation model based on D-JEPA. The results indicate that D-JEPA can generate continuous data and be used to build multimodal understanding and generation models (Appendix G).

## 5 CONCLUSION

In this study, we have introduced D-JEPA as a novel generative model that can handle the generation of continuous data. Through extensive experiments, we have preliminarily validated the feasibility of this approach. However, scaling up the D-JEPA model remains a significant challenge that warrants further investigation. Additionally, the acceleration strategies and application ecosystems for diffusion models and auto-regressive models have already reached a high level of maturity. Adopting these successful strategies for the further development of D-JEPA is crucial. We hope that this work will inspire future researchers to explore the D-JEPA architecture, laying the groundwork for the construction of a unified multimodal model in the future.

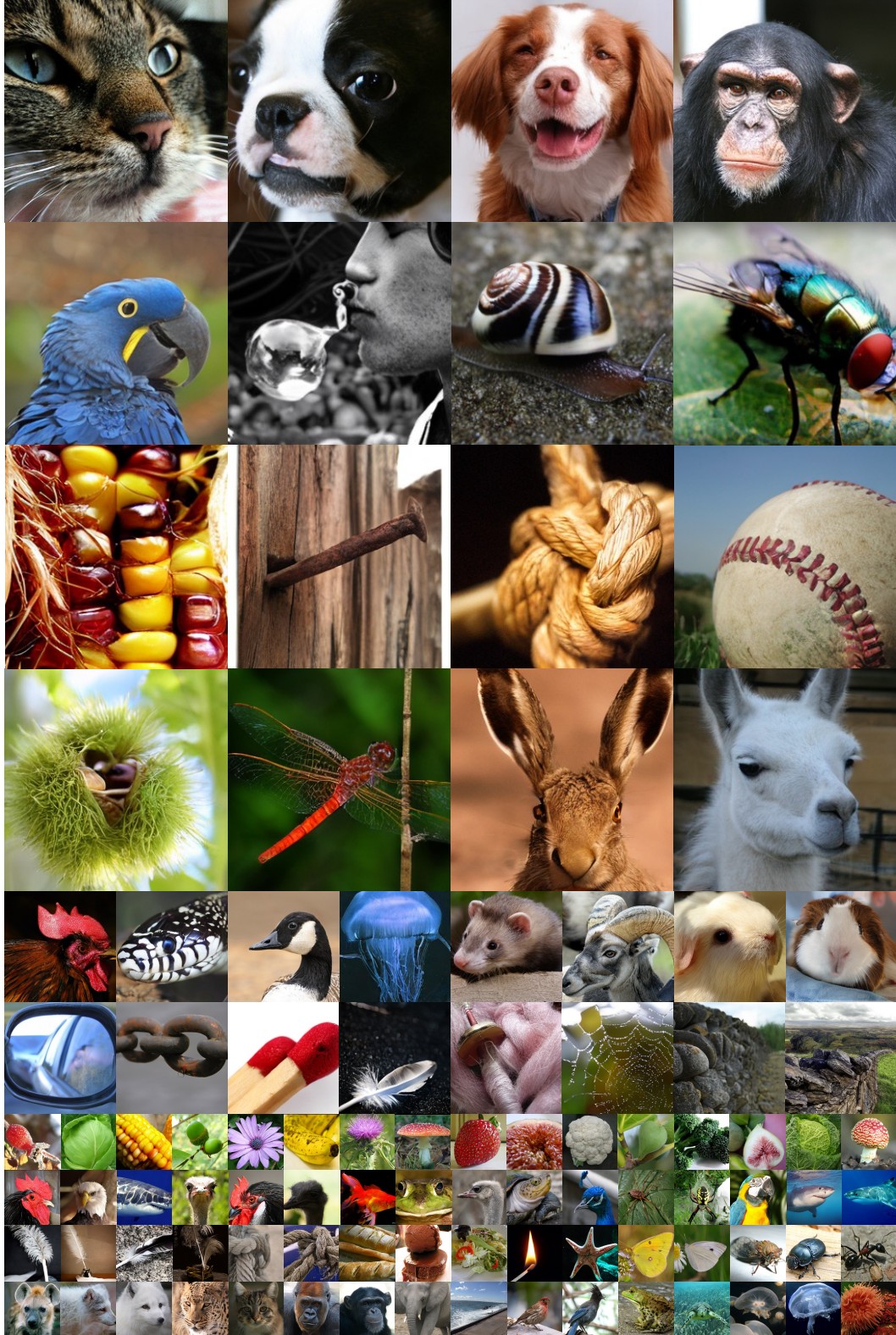

Figure 3: D-JEPA achieves state-of-the-art image quality. We showcase selected high-fidelity examples of class-conditional generation on ImageNet $256 \times 256$ using D-JEPA-H.

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

## A   RELATED WORK

**Self-supervised learning.**   Early efforts in unsupervised learning emphasized pretext tasks designed to predict pseudo labels, such as solving jigsaw puzzles (Noroozi & Favaro, 2016), restoring missing patches (Pathak et al., 2016), and predicting image rotations (Gidaris et al., 2018). Despite yielding useful representations, these methods lagged behind those acquired through supervised learning.

The advent of contrastive learning marked a significant leap forward, achieving near-supervised performance with techniques like MoCo (He et al., 2020) and contrastive multiview coding (Tian et al., 2020). BYOL (Grill et al., 2020) further advanced the field by eliminating the need for negative pairs and employing peer networks for mutual learning. However, these methods primarily focused on image data and could not be directly applied to other continuous data types, such as audio.

Inspired by developments in natural language processing, masked image modeling (MIM) emerged as an effective approach to self-supervised learning (Kenton & Toutanova, 2019), extendable to continuous data types such as audio (Baevski et al., 2022). Models such as BEiT (Bao et al., 2021), PeCo (Dong et al., 2021), and MAE He et al. (2022a) focus on recovering visual tokens or performing pixel-level reconstruction. DiffMAE (Wei et al., 2023) leverages denoising diffusion decoders primarily for representation learning rather than image generation. While MIM-based methods prioritize downstream performance, they often result in suboptimal reconstructions (He et al., 2022a; Bao et al., 2021).

Recent advances such as MAGE (Li et al., 2023) achieve high-quality representations and image generation. However, these methods are limited to image data, and have not been validated on other types of continuous data such as speech and video. Our work uniquely demonstrates the significant impact of representation learning across image, audio, and video generation, highlighting its versatility and effectiveness in these domains.

**Generative modeling.**   Generative models have seen significant advances in image synthesis. Generative adversarial networks (GANs) (Goodfellow et al., 2014; Zhang et al., 2017a; Karras et al., 2019; Zhang et al., 2019; Brock et al., 2019) excel in producing realistic images but often face challenges such as training instability and mode collapse. Two-stage models (Van Den Oord et al., 2017; Chang et al., 2022; Yu et al., 2021; Lee et al., 2022) tokenize images and apply maximum likelihood estimation in the latent space, with VQ-VAE-2 (Razavi et al., 2019) generating diverse samples. ViT-VQGAN (Yu et al., 2021) utilizes vision transformers (Dosovitskiy et al., 2020), while MaskGIT (Chang et al., 2022) employs bidirectional transformers for faster decoding. Additionally, diffusion models (Ho et al., 2020; Song et al., 2021; Dhariwal & Nichol, 2021; Rombach et al., 2022) have achieved remarkable results in image synthesis.

Autoregressive models (Gregor et al., 2014; Van Den Oord et al., 2016; Van den Oord et al., 2016; Parmar et al., 2018; Chen et al., 2018; 2020) generate images as pixel sequences, utilizing recurrent neural networks (Van Den Oord et al., 2016), convolutional neural networks  (Van den Oord et al., 2016; Chen et al., 2018), and transformers  (Parmar et al., 2018; Chen et al., 2020). Recent approaches  (Van Den Oord et al., 2017; Razavi et al., 2019; Esser et al., 2021; Ramesh et al., 2021) adopt discrete tokens inspired by language models for image modeling. However, training these discrete tokenizers is challenging for high-quality image generation (Kolesnikov et al., 2022; Yu et al., 2023a; Mentzer et al., 2023).

Recent developments, such as GIVT (Tschannen et al., 2023) and MAR (Li et al., 2024), introduce continuous-valued tokens, yielding promising results. Our method directly processes continuous-valued tokens via the diffusion process, ensuring high-quality image generation.

## B   EXPERIMENTAL SETUP

**Network configuration.**   We constructed three variants of D-JEPA using different visual transformer backbones: D-JEPA-B with ViT-B, D-JEPA-L with ViT-L, and D-JEPA-H with ViT-H. In all configurations, $u_\theta$ is implemented as a simple two-layer MLP. The denoising MLP, denoted as $\epsilon_\theta$, follows the implementation described in Li et al. (2024), comprising 6, 8, and 12 residual blocks with linear layers, respectively.

We adhered to the standard practice outlined in LeCun (2022) for parameter counting, considering only the learnable parameters updated through gradients. Since the target encoder $\bar{\phi}$ is updated via exponential moving average and is not utilized during sampling, its parameters are excluded from the count. The number of parameters for each model is detailed in Tab. 1.

**Training.**    We trained class-conditional latent D-JEPA models at a $256 \times 256$ image resolution on the ImageNet dataset (Krizhevsky et al., 2012), a highly competitive benchmark for generative modeling. The VAE trained by Li et al. (2024) encodes images into a latent space of size $16 \times 32 \times 32$. Subsequently, we employ a patch size of $p = 1$ to segment the latent space into semantic tokens, resulting in $N = 256$ tokens. After sampling all tokens with D-JEPA according to Algorithm 1, we decode them to pixels using the corresponding VAE decoder. We retain the diffusion hyperparameters from ADM (Dhariwal & Nichol, 2021); specifically, we use a $t_{max} = 1000$ linear variance schedule ranging from $1 \times 10^{-4}$ to $2 \times 10^{-2}$, ADM's parameterization of the covariance $\Sigma_\theta$, and their method for embedding input timesteps and labels.

The final linear layer is initialized with zeros, while other layers follow standard weight initialization techniques from ViT (Dosovitskiy et al., 2020). We train all models using AdamW (Loshchilov & Hutter, 2017) with a learning rate of $8 \times 10^{-4}$, incorporating a 100-epoch linear warmup and a linear weight decay from 0.02 to 0.2 for all parameters except $u_\theta$, $\epsilon_\theta$, and $\bar{\phi}$. The experiments are conducted on four workers, each equipped with 8 H800 GPUs, with a total batch size 2048. The only data augmentation applied is horizontal flipping.

Following standard practices in generative modeling, we maintain an exponential moving average of D-JEPA weights throughout training, with a decay rate of 0.9999. All reported results are based on the EMA model. The number of training epochs varies depending on the model's performance during evaluation; training is halted once the evaluation metrics show no significant improvement.

**Evaluation metrics.**    We measure scaling performance using the Fréchet Inception Distance (FID) (Heusel et al., 2017), the standard metric for evaluating generative models of images. In line with convention, we compare against prior works and report FID-50K using 100 DDPM sampling steps (Ho et al., 2020). FID is known to be sensitive to small implementation details (Parmar et al., 2022); to ensure accurate comparisons, all values reported in this paper are obtained by exporting samples and using ADM's evaluation suite (Dhariwal & Nichol, 2021). FID numbers reported in this paper do not use classifier-free guidance except where otherwise stated. Additionally, we report the Inception Score (IS) (Salimans et al., 2016) and Precision/Recall (Kynkäänniemi et al., 2019) as secondary metrics.

## C    SAMPLING WITH GENERALIZED NEXT TOKEN PREDICTION

### C.1    GRID SEARCHING FOR OPTIMAL CLASSIFIER-FREE GUIDANCE SCALE AND TEMPERATURE

As described in Sec. 3.3, the temperature $\tau$ controls the noise level added at each iteration during the denoising process and significantly influences the final sampling results. Therefore, we conducted a grid search to determine each model's optimal $\tau$. Similarly, the classifier-free guidance scale cfg is known to affect sampling results notably and was also included in the grid search. We use 100 DDPM sampling steps for the denoising MLP following Li et al. (2024); Ho et al. (2020) and set the auto-regressive steps to 64, which balances sampling efficiency and image quality.

**Coarse searching.**    In the coarse searching stage, we began by exploring $\tau$ from 0.90 to 1.05 with a step size of 0.01, considering cfg $\in \{1.0, 2.0, 3.0, 4.0\}$. The results are plotted in Fig. 4a, Fig. 4c, and Fig. 4e.

**Fine searching.**    Based on cfg $= 3.0, 4.0$ and $\tau = 0.97$, we conducted a fine search for the best $\text{cfg}_c - \tau_c$ combinations. In this stage, a 2D grid search was performed with cfg ranging from $[\text{cfg}_c - 0.3, \text{cfg}_c + 0.3]$ with a step size of 0.1, and $\tau_c$ ranging from $[\tau_c - 0.01, \tau_c + 0.01]$ with a step size of 0.01, as illustrated in Fig. 4b, Fig. 4c, and Fig. 4f.

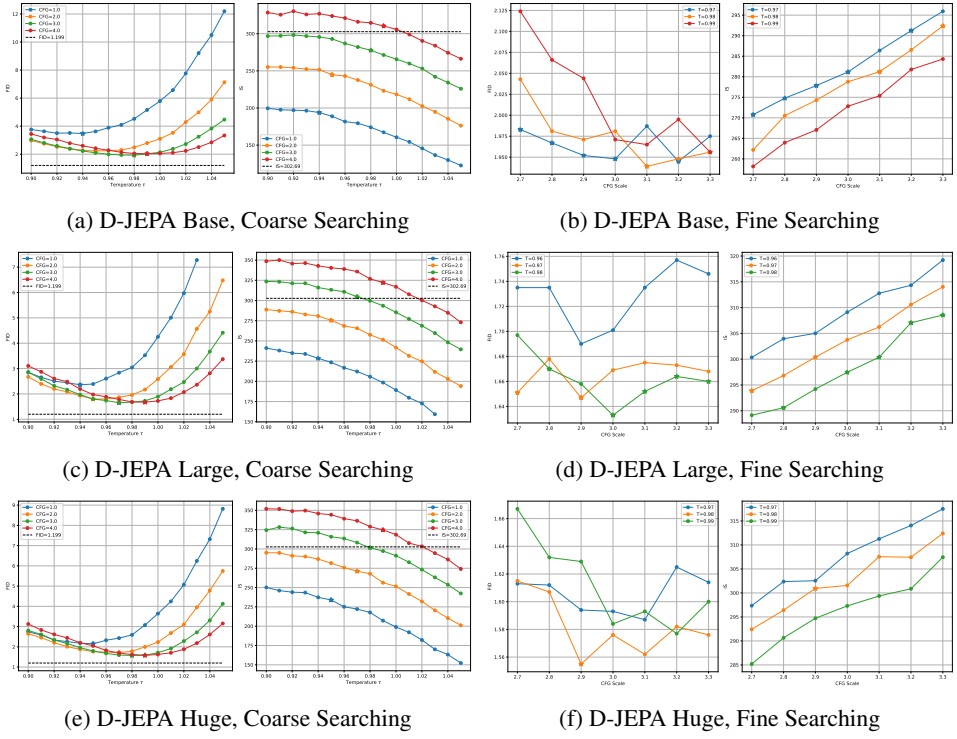

Figure 4: Grid searching for CFG scale and temperature $\tau$. FID=1.199 and IS=302.69 represent the benchmarks achievable by VAE, which are the theoretical upper limits for D-JEPA.

When cfg $= 1.0$, *i.e.*, no classifier-free guidance, we found that $\tau$ in the range $[0.94, 0.95]$ works well for all models. For cfg $\geq 1.0$, $\tau$ in the range $[0.97, 0.99]$ performs better for all models. Fig. 4 also indicates that cfg $= 3.0$ can achieve a lower FID, while cfg $= 4.0$ can achieve a better IS. Therefore, we recommend setting $\tau$ to 0.98 for classifier-free guidance and cfg to either 3.0 or 4.0 as the default.

## C.2 ABALATION ON AUTO-REGRESSIVE STEPS

Unlike the typical next token prediction that predicts only the next token in raster order, the generalized next token prediction can predict multiple tokens in random order, referred to as the next set-of-token prediction in Li et al. (2024). As mentioned in Sec. 3.5, we use an iterative sampling strategy with a cosine step schedule. The auto-regressive steps $N$ directly affect image quality and sampling efficiency.

We started with $T = 32$ and incrementally increased by 32 to 256. Note that at $T = 256$, the next set-of-tokens prediction decays to the typical next token prediction. We conducted experiments both with and without classifier-free guidance. The former requires additional null condition inference, thus doubling the computational cost.

The results are summarized in Tab. 3. We found that sampling with more than 64 steps for models across all scales significantly outperforms 32 steps. We also visualize the samples in Fig. 5. Notably, at 64 steps, the sampled images are already very close to the optimal results more efficiently. Contrary to our initial assumptions, we discovered that *achieving the best sampling performance does not necessarily require 256 steps.* This highlights the advantage of the next set-of-token prediction over typical next-token prediction in terms of sampling efficiency.

Another interesting observation is that the steps required to achieve the best FID consistently decrease as the model size increases. Specifically, we achieve the best performance for the huge model with only 64 steps.

| AR Steps | 32 | 64 | 96 | 128 | 160 | 192 | 224 | 256 |
|---|---|---|---|---|---|---|---|---|
| FID↓ | 4.027 | 3.452 | 3.431 | **3.404** | 3.419 | 3.467 | 3.437 | 3.421 |
| IS↑ | 184.42 | 194.57 | 195.74 | 197.07 | 196.00 | 195.29 | 194.79 | 196.03 |
| Pre.↑ | 0.756 | 0.764 | 0.764 | 0.766 | 0.767 | 0.763 | 0.765 | 0.763 |
| Rec.↑ | 0.614 | 0.613 | 0.602 | 0.608 | 0.609 | 0.609 | 0.604 | 0.607 |
| Sec/Img | 0.043 | 0.078 | 0.114 | 0.150 | 0.189 | 0.227 | 0.261 | 0.294 |
| AR Steps | 32 | 64 | 96 | 128 | 160 | 192 | 224 | 256 |
| FID↓ | 2.261 | 1.912 | 1.896 | 1.885 | 1.885 | 1.904 | **1.871** | 1.881 |
| IS↑ | 271.57 | 281.46 | 281.28 | 280.78 | 282.38 | 280.15 | 282.53 | 283.92 |
| Pre.↑ | 0.785 | 0.800 | 0.797 | 0.798 | 0.799 | 0.802 | 0.797 | 0.802 |
| Rec.↑ | 0.603 | 0.603 | 0.609 | 0.611 | 0.614 | 0.609 | 0.605 | 0.613 |
| Sec/Img | 0.069 | 0.120 | 0.182 | 0.231 | 0.309 | 0.381 | 0.411 | 0.539 |
| AR Steps | 32 | 64 | 96 | 128 | 160 | 192 | 224 | 256 |
| FID↓ | 2.143 | **2.081** | 2.118 | 2.139 | 2.139 | 2.183 | 2.096 | 2.149 |
| IS↑ | 313.57 | 320.94 | 318.96 | 318.67 | 318.94 | 318.28 | 317.93 | 320.62 |
| Pre.↑ | 0.810 | 0.817 | 0.815 | 0.819 | 0.819 | 0.820 | 0.818 | 0.820 |
| Rec.↑ | 0.586 | 0.590 | 0.589 | 0.595 | 0.588 | 0.589 | 0.592 | 0.590 |
| Sec/Img | 0.069 | 0.120 | 0.182 | 0.231 | 0.309 | 0.381 | 0.411 | 0.539 |
| AR Steps | 32 | 64 | 96 | 128 | 160 | 192 | 224 | 256 |
| FID↓ | 2.996 | 2.371 | 2.344 | 2.344 | **2.323** | 2.347 | 2.345 | 2.362 |
| IS↑ | 211.85 | 228.99 | 231.36 | 231.83 | 233.46 | 229.20 | 230.54 | 231.21 |
| Pre.↑ | 0.765 | 0.780 | 0.781 | 0.783 | 0.785 | 0.782 | 0.786 | 0.785 |
| Rec.↑ | 0.620 | 0.626 | 0.623 | 0.617 | 0.616 | 0.618 | 0.613 | 0.617 |
| Sec/Img | 0.079 | 0.146 | 0.218 | 0.293 | 0.363 | 0.435 | 0.554 | 0.585 |
| AR Steps | 32 | 64 | 96 | 128 | 160 | 192 | 224 | 256 |
| FID↓ | 2.042 | 1.633 | 1.651 | 1.620 | 1.637 | 1.600 | 1.639 | **1.593** |
| IS↑ | 282.91 | 297.46 | 300.76 | 299.95 | 300.16 | 300.96 | 302.28 | 303.13 |
| Pre.↑ | 0.783 | 0.796 | 0.802 | 0.799 | 0.799 | 0.800 | 0.803 | 0.801 |
| Rec.↑ | 0.623 | 0.627 | 0.623 | 0.619 | 0.623 | 0.623 | 0.621 | 0.613 |
| Sec/Img | 0.139 | 0.248 | 0.361 | 0.481 | 0.601 | 0.743 | 0.859 | 1.003 |
| AR Steps | 32 | 64 | 96 | 128 | 160 | 192 | 224 | 256 |
| FID↓ | 1.859 | 1.663 | 1.706 | 1.700 | 1.690 | **1.649** | 1.714 | 1.674 |
| IS↑ | 313.27 | 325.78 | 325.52 | 329.41 | 326.76 | 327.20 | 330.74 | 328.52 |
| Pre.↑ | 0.796 | 0.807 | 0.813 | 0.810 | 0.809 | 0.810 | 0.811 | 0.813 |
| Rec.↑ | 0.614 | 0.617 | 0.612 | 0.612 | 0.617 | 0.614 | 0.605 | 0.609 |
| Sec/Img | 0.139 | 0.248 | 0.362 | 0.481 | 0.604 | 0.747 | 0.862 | 1.003 |
| AR Steps | 32 | 64 | 96 | 128 | 160 | 192 | 224 | 256 |
| FID↓ | 2.849 | 2.174 | 2.106 | 2.083 | 2.157 | **2.043** | 2.133 | 2.112 |
| IS↑ | 213.62 | 232.50 | 235.90 | 239.18 | 236.30 | 239.27 | 236.90 | 237.74 |
| Pre.↑ | 0.766 | 0.783 | 0.783 | 0.784 | 0.787 | 0.785 | 0.783 | 0.783 |
| Rec.↑ | 0.634 | 0.621 | 0.622 | 0.627 | 0.623 | 0.620 | 0.630 | 0.625 |
| Sec/Img | 0.128 | 0.237 | 0.348 | 0.463 | 0.585 | 0.705 | 0.836 | 0.969 |
| AR Steps | 32 | 64 | 96 | 128 | 160 | 192 | 224 | 256 |
| FID↓ | 1.859 | 1.551 | 1.567 | **1.542** | 1.558 | 1.574 | 1.901 | 1.573 |
| IS↑ | 307.07 | 322.79 | 325.04 | 324.17 | 326.59 | 321.93 | 320.48 | 326.27 |
| Pre.↑ | 0.787 | 0.802 | 0.807 | 0.806 | 0.808 | 0.806 | 0.804 | 0.807 |
| Rec.↑ | 0.620 | 0.622 | 0.622 | 0.619 | 0.619 | 0.616 | 0.615 | 0.620 |
| Sec/Img | 0.235 | 0.420 | 0.606 | 0.800 | 0.994 | 1.197 | 1.417 | 1.659 |
| AR Steps | 32 | 64 | 96 | 128 | 160 | 192 | 224 | 256 |
| FID↓ | 1.718 | **1.661** | 1.669 | 1.669 | 1.706 | 1.703 | 1.680 | 1.726 |
| IS↑ | 325.62 | 339.62 | 339.00 | 339.71 | 340.75 | 338.80 | 341.01 | 340.00 |
| Pre.↑ | 0.800 | 0.814 | 0.816 | 0.816 | 0.815 | 0.816 | 0.816 | 0.817 |
| Rec.↑ | 0.610 | 0.614 | 0.614 | 0.608 | 0.610 | 0.604 | 0.604 | 0.611 |
| Sec/Img | 0.235 | 0.420 | 0.606 | 0.800 | 0.994 | 1.197 | 1.417 | 1.659 |

Table 3: Ablation in auto-regressive steps. (cfg, $\tau$) combinations: Base, $(1.0, 0.93)$, $(3.1, 0.98)$, $(4.1, 0.97)$; Large, $(1.0, 0.94)$, $(3.0, 0.98)$, $(3.9, 0.98)$; Huge, $(1.0, 0.95)$, $(3.9, 0.99)$, $(4.3, 0.98)$.

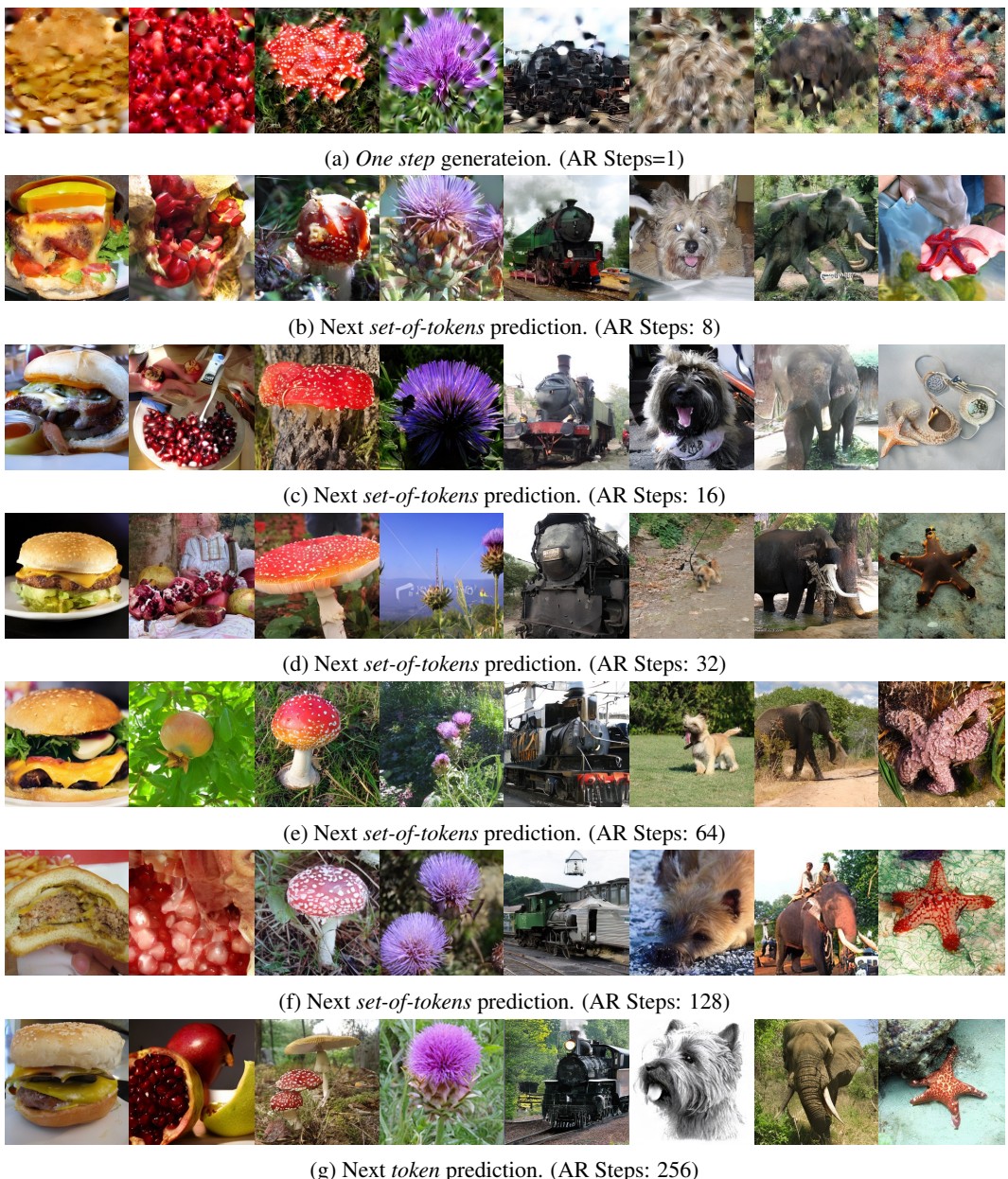

(a) *One step* generateion. (AR Steps=1)

(b) Next *set-of-tokens* prediction. (AR Steps: 8)

(c) Next *set-of-tokens* prediction. (AR Steps: 16)

(d) Next *set-of-tokens* prediction. (AR Steps: 32)

(e) Next *set-of-tokens* prediction. (AR Steps: 64)

(f) Next *set-of-tokens* prediction. (AR Steps: 128)

(g) Next *token* prediction. (AR Steps: 256)

Figure 5: **Uncurated** $256 \times 256$ D-JEPA-H samples with different auto-regressive steps. cfg $= 3.9, \tau = 0.99$.

**Sampling efficiency.** We measured sampling efficiency using H800 hardware with a batch size 256, averaging the time to generate one image with a resolution of $256 \times 256$. Results under selected configurations are plotted in Fig. 2, and more details are listed in Tab. 4.

D-JEPA can generate images with an FID of 4.0 in 47 ms, demonstrating strong competitiveness. Moreover, we can achieve state-of-the-art FID within 1 second. In Tab. 5, we list the configuration used in Sec. 4.1.

| Model | cfg | $\tau$ | AR Steps | FID↓ | IS↑ | Pre.↑ | Rec.↑ | Sec/Img |
|-------|-----|--------|----------|------|-----|-------|-------|---------|
| Base | 1.0 | 0.93 | 32 | 4.027 | 184.42 | 0.756 | 0.614 | 0.043 |
| Base | 3.1 | 0.98 | 32 | 2.261 | 271.57 | 0.785 | 0.603 | 0.069 |
| Base | 4.1 | 0.97 | 64 | 2.081 | 320.94 | 0.817 | 0.590 | 0.120 |
| Base | 3.1 | 0.98 | 64 | 1.912 | 281.46 | 0.800 | 0.603 | 0.120 |
| Large | 3.0 | 0.98 | 64 | 1.633 | 297.46 | 0.796 | 0.627 | 0.248 |
| Huge | 4.3 | 0.98 | 64 | 1.661 | 339.62 | 0.814 | 0.614 | 0.463 |
| Huge | 3.9 | 0.99 | 128 | 1.542 | 324.17 | 0.806 | 0.619 | 0.800 |

Table 4: Details of efficiency configuration. **Sec/Img** denotes the average time to generate one image (measured with a batch size of 256 on an H800 GPU).

| | w/o cfg | | w/ cfg (best FID) | | | w/ cfg (better IS) | | |
|-------|---------|--------|-----|----------|--------|-----|----------|--------|
| Model | AR Steps | $\tau$ | cfg | AR Steps | $\tau$ | cfg | AR Steps | $\tau$ |
| D-JEPA-B | 128 | 0.93 | 3.1 | 224 | 0.98 | 4.1 | 64 | 0.97 |
| D-JEPA-L | 160 | 0.94 | 3.0 | 256 | 0.98 | 3.9 | 192 | 0.98 |
| D-JEPA-H | 192 | 0.95 | 3.9 | 128 | 0.99 | 4.3 | 224 | 0.98 |

Table 5: The sampling configuration for results listed in Tab. 2, Tab. 1 and Tab. 7.

### C.3 ABLATION ON JEPA LOSS

We conducted ablation experiments on the D-JEPA-B model. The results of MAR-B serve as the baseline (*i.e.*, the results without the JEPA Loss), because MAR-B and D-JEPA-B have identical model structures and experimental parameters, making them a comparable baseline.

From Fig. 6, we can see that D-JEPA-B achieves the performance of MAR-B at 600 epochs. Additionally, from Tab. 1 and Tab. 2, we observe that D-JEPA-B significantly outperforms MAR-B at 1400 epochs. For a more intuitive comparison, we provide some key data points without using classifier-free guidance in Tab. 6, which we believe better reflects the true scenario.

## D ADDITIONAL RESULTS ON IMAGENET

### D.1 FULL COMPARISON ON IMAGENET

We provide a more comprehensive comparison with earlier works in Tab. 7. D-JEPA-B, with only 212M parameters, performs only slightly worse than MAR-L (with 479M parameters), MAR-H (with 943M parameters), and VDM++ (with 2.0B parameters). It is also worth noting that D-JEPA-B significantly outperforms DiT-XL (with 675M parameters) by a large margin in terms of FID (3.40 vs. 9.62) and IS (197.1 vs. 121.5).

D-JEPA-L and D-JEPA-H achieve state-of-the-art performance in both FID and IS metrics. Furthermore, D-JEPA-H sets unprecedented records in both of these metrics. Although D-JEPA performs well in Tab. 2, it is not outstanding. However, we believe that the metrics without classifier-free guidance better reflect the actual performance of each model, as the value of cfg can significantly impact the results. Determining the optimal cfg value for each model is challenging. Therefore, the metrics presented in Tab. 7 are considered more convincing.

| Model | #params | Epoch | FID(w/o cfg) | IS(w/o cfg) | FID(w/ cfg) | IS(w/ cfg) |
|-------|---------|-------|--------------|-------------|-------------|------------|
| MAR-B | 208M | 800 | 3.48 | 192.4 | 2.31 | 281.7 |
| D-JEPA-B | 208M+4M | 600 | 3.50 | 189.2 | 2.25 | 279.8 |
| | | 800 | 3.43 | 195.3 | 2.01 | 280.1 |
| | | 1400 | 3.40 | 197.1 | 1.87 | 282.5 |

Table 6: Ablation on JEPA loss. The additional 4M parameters are introduced by the MLP layers in JEPA loss. CFG is set as 3.1 for D-JEPA-B.

| | #Params | FID↓ | IS↑ | Pre.↑ | Rec.↑ |
|---|---|---|---|---|---|
| *Base scale model (less than 300M)* | | | | | |
| MAR-B (2024) | 208M | 3.48 | 192.4 | 0.78 | 0.58 |
| D-JEPA-B | 212M | **3.40** | **197.1** | 0.77 | **0.61** |
| MaskGIT (2022) | 227M | 6.18 | 182.1 | **0.80** | 0.51 |
| MAGE (2023) | 230M | 6.93 | 195.8 | - | - |
| *Large scale model (300∼700M)* | | | | | |
| GIVT (2023) | 304M | 5.67 | - | 0.75 | 0.59 |
| MAGVIT-v2 (2023b) | 307M | 3.65 | 200.5 | - | - |
| LDM-4 (2022) | 400M | 10.56 | 103.5 | 0.71 | 0.62 |
| MAR-L (2024) | 479M | 2.60 | 221.4 | **0.79** | 0.60 |
| ADM (2021) | 554M | 10.94 | 101.0 | 0.63 | 0.63 |
| DiT-XL (2023) | 675M | 9.62 | 121.5 | 0.67 | **0.67** |
| SiT-XL (2024a) | 675M | 8.60 | - | - | - |
| MDTv2-XL (2023) | 676M | 5.06 | 155.6 | 0.72 | 0.66 |
| D-JEPA-L | 687M | **2.32** | **233.5** | **0.79** | 0.62 |
| *Huge scale model (900+M)* | | | | | |
| MAR-H (2024) | 943M | 2.35 | 227.8 | **0.79** | **0.62** |
| D-JEPA-H | 1.4B | **2.04** | **239.3** | **0.79** | **0.62** |
| Autoreg. (2021) | 1.4B | 15.78 | 78.3 | - | - |
| VDM++ (2024) | 2.0B | 2.40 | 225.3 | - | - |

Table 7: System-level comparison on ImageNet $256 \times 256$ conditional generation *without* classifier-free guidance.

| | | w/o CFG | | w/ CFG | |
|---|---|---|---|---|---|
| | #params | FID↓ | IS↑ | FID↓ | IS↑ |
| *pixel-based* | | | | | |
| ADM (2021) | 554M | 23.24 | 58.1 | 7.72 | 172.7 |
| VDM++ (2024) | 2B | 2.99 | 232.2 | 2.65 | 278.1 |
| *vector-quantized tokens* | | | | | |
| MaskGIT (2022) | 227M | 7.32 | 156.0 | - | - |
| MAGVIT-v2 (2023b) | 307M | 3.07 | 213.1 | 1.91 | 324.3 |
| *continuous-valued tokens* | | | | | |
| U-ViT-H/2-G (2022) | 501M | - | - | 4.05 | 263.8 |
| DiT-XL/2 (2023) | 675M | 12.03 | 105.3 | 3.04 | 240.8 |
| DiffiT (2025) | - | - | - | 2.67 | 252.1 |
| GIVT (2025) | 304M | 8.35 | - | - | - |
| EDM2-XXL (2024) | 1.5B | 1.91 | - | 1.81 | - |
| MAR-L (2024) | 481M | 2.74 | 205.2 | 1.73 | 279.9 |
| D-JEPA-L | 687M | **1.89** | **239.1** | **1.62** | **335.2** |

Table 8: System-level comparison on ImageNet $512 \times 512$ conditional generation. D-JEPA-L is trained for 480 epochs.

## D.2 ADDITIONAL COMPARISON ON IMAGENET $512 \times 512$

Following previous works, we also report results on ImageNet at a resolution of $512 \times 512$, compared with leading systems (see Tab. 8). For simplicity, we use the KL-16 tokenizer, which results in a sequence length of $32 \times 32$ for a $512 \times 512$ image. Other settings follow the D-JEPA-L configuration. Our method achieves a Fréchet Inception Distance (FID) of 1.89 without Classifier-Free Guidance (CFG) and 1.62 with CFG. Our results surpass those of previous systems. Due to limited resources, we have not trained the larger D-JEPA-H model on ImageNet $512 \times 512$, which is expected to yield even better results.

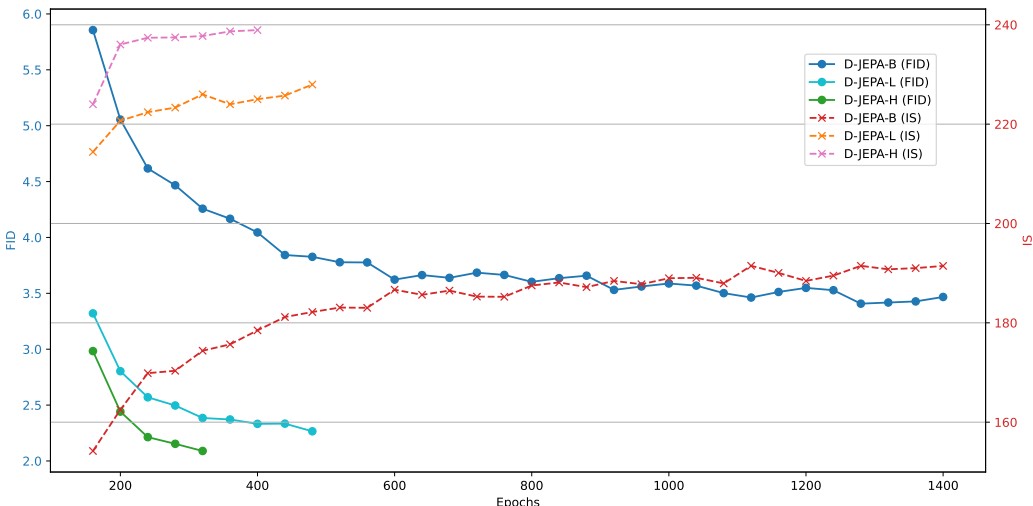

Figure 6: Training dynamics of model architectures. The sampling parameters for the images evaluated to obtain the FID and IS in the figure are: cfg $= 1.0, \tau = 0.95$, AR Steps $= 256$.

### D.3 TRAINING CURVE

In Fig. 6, we present the FID and IS metrics as a function of training epochs. It is evident that irrespective of the model size, 160 training epochs are essential for convergence when employing diffusion loss. Models with a smaller parameter count necessitate a more significant number of training epochs. Although the Base, Large, and Huge models were trained for 1400, 480, and 320 epochs, respectively, the trends in the metrics suggest that further increasing the number of training epochs could potentially enhance D-JEPA's performance.

Moreover, it is crucial to highlight that the model size directly influences the attainable performance ceiling. For smaller models, such as D-JEPA-B, achieving the performance levels of larger models, like D-JEPA-L and D-JEPA-H, is unattainable, even with extensive training. As illustrated in Fig. 6, it is apparent that when the parameter size exceeds 500M, D-JEPA exhibits markedly superior learning capabilities.

### D.4 INPAINTING AND OUTPAINTING

In Fig. 7 and Fig. 8, we demonstrate D-JEPA's capabilities in image restoration and inpainting and outpainting on both ImageNet1k images and unseen images. As shown in Fig. 7, D-JEPA can reconstruct highly similar images from only 10% of the original features, showcasing its visual solid retrieval capabilities. In Fig. 8, D-JEPA effectively supplements missing visual features based on contextual content.

We believe that training on larger-scale datasets will significantly enhance D-JEPA's painting capabilities. Additionally, from the results in these two figures, it is evident that the quality of the training data plays a decisive role in the final generated image quality. D-JEPA, like all other generative models, essentially strives to fit the training data distribution better and finds it challenging to create data distributions that are perceived as better by human standards.

### E D-JEPA FOR REPRESENTATION LEARNING

#### E.1 THEORETICAL MOTIVATION

A theoretical motivation for the effectiveness of the collapse prevention strategy was proposed in Grill et al. (2020). We adapt their analysis for our smoothed l1 loss in Eq. 2. For ease of exposition, we will disregard the effect of the conditioning variable and consider one-dimensional representa-

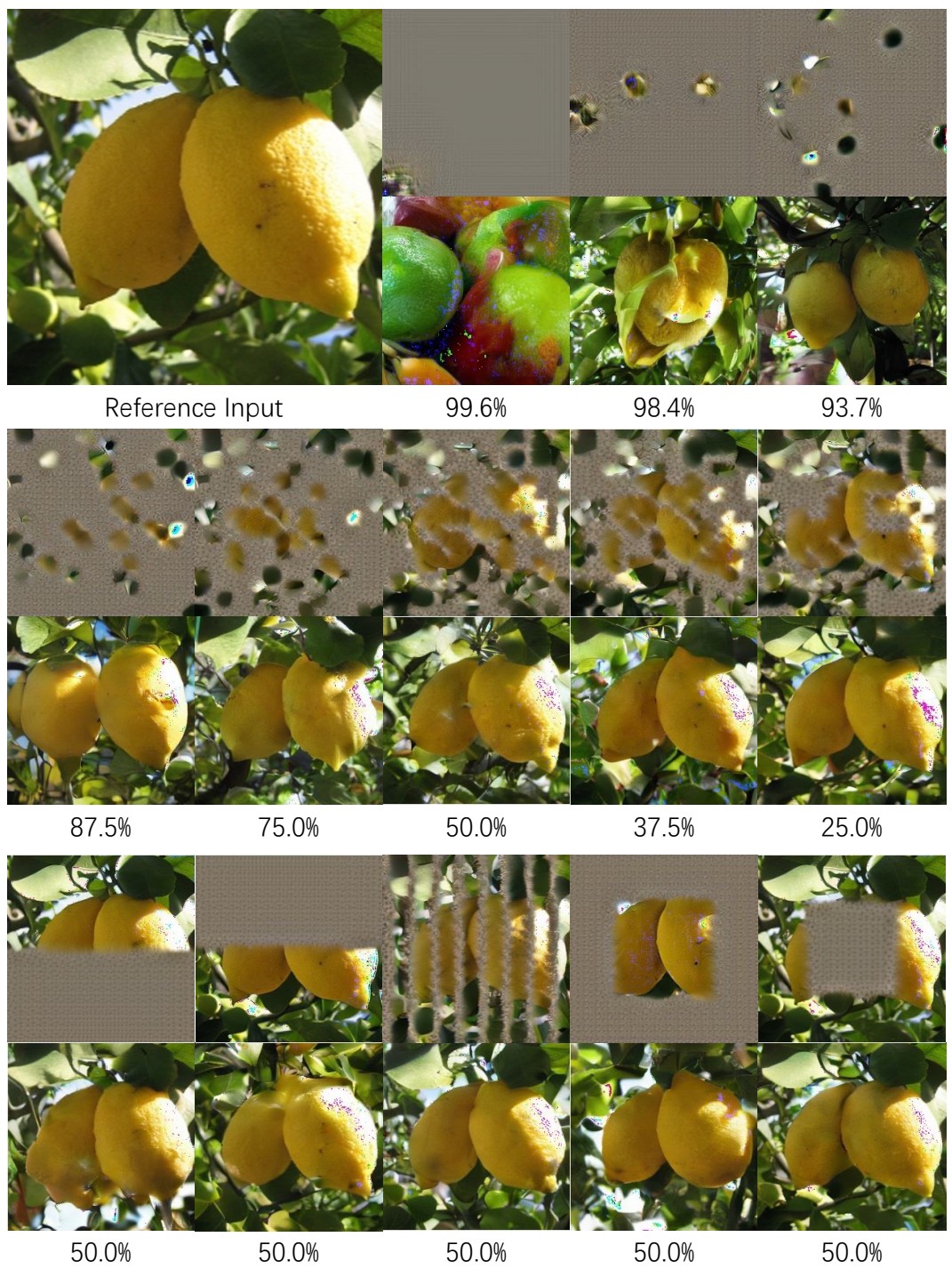

Figure 7: Inpainting and outpainting capabilities of D-JEPA on training data. The percentages below the images indicate the proportion of the reference input that was dropped.

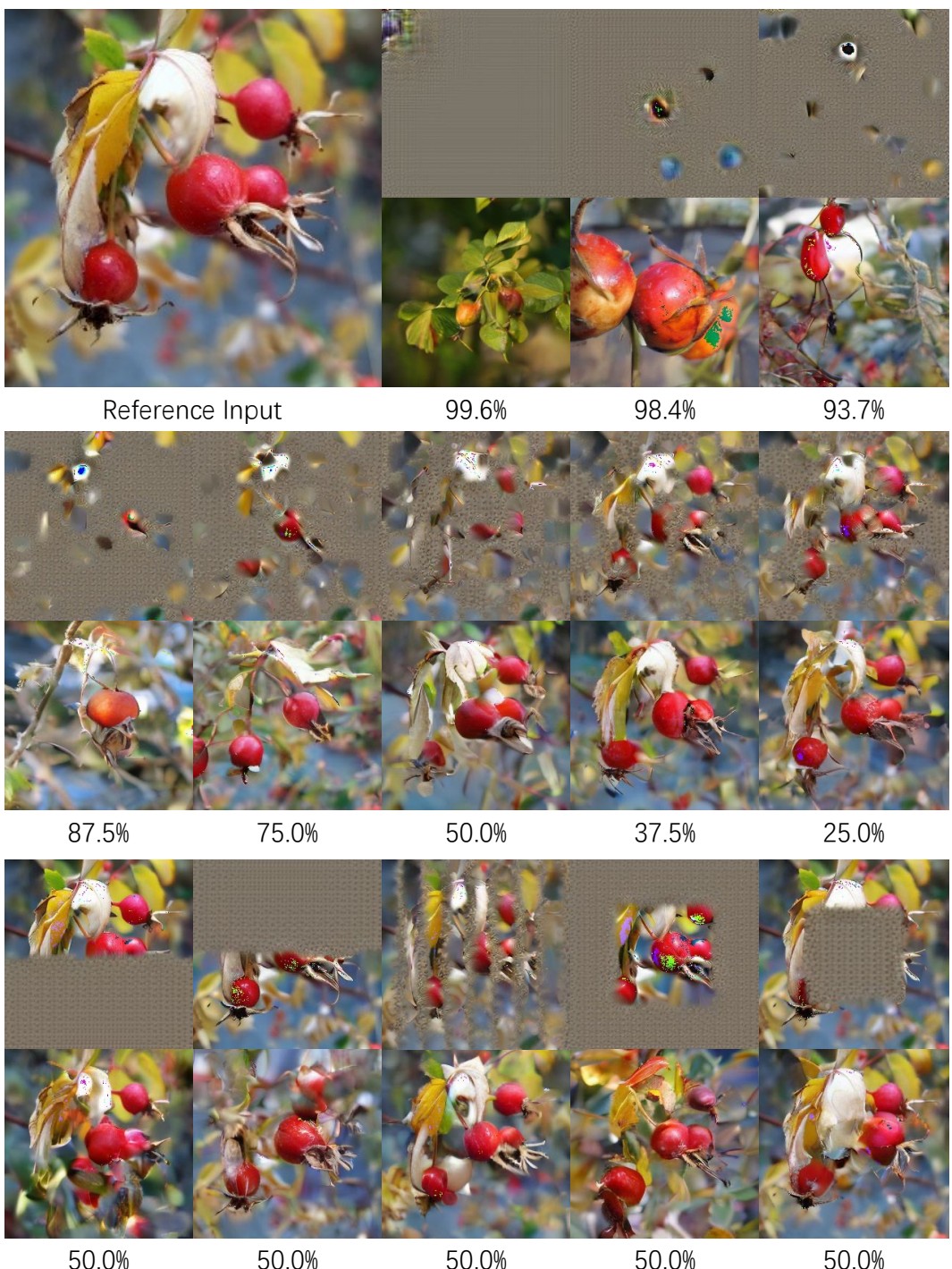

Figure 8: Inpainting and outpainting capabilities of D-JEPA on previously unseen data. The percentages below the images indicate the proportion of the reference input that was dropped.

tions like Bardes et al. (2024). The optimal predictor under Eq. 2 is thus given by the following functional expression:

$$\gamma^{\star}(\phi(x_i)) = \text{argmin}_{\gamma} \parallel \gamma(\phi(x_i)) - g_i \parallel_1$$
$$= \text{media}(g_i | \phi(x_i)),$$

Substituting this expression for the optimal predictor into the loss function and evaluating the expected gradient of the encoder gives

$$\nabla_{\theta} \mathbb{E} \parallel \gamma^{\star}(\phi_{\theta}(x_i)) - g_i \parallel_1 = \nabla_{\theta} \text{MAD}(g_i | \phi_{\theta}(x_i)),$$

where $\text{MAD}(\cdot | \phi_{\theta}(x_i))$ is the median absolute deviation of a random variable conditioned on $\phi_{\theta}(x_i)$. Thus, in the case where the predictor is optimal, the encoder must learn to capture as much information about the video as possible to minimize the deviation of the target. The hypothesis is that incorporating an exponential moving average to compute the representation of $y_i$ ensures that the predictor evolves faster than the encoder and remains close to optimal, thereby preventing collapse.

### E.2 IMAGE CLASSIFICATION

D-JEPA, a generative model built on the foundation of JEPA LeCun (2022), inherits the exceptional representation capabilities of JEPA. This intrinsic quality positions D-JEPA as a potentially superior source of pre-training weights for downstream tasks compared to other generative models. To rigorously affirm this hypothesis, we conducted extensive evaluations on two prevalent ImageNet classification tasks: linear probing and fine-tuning.

**Experiment settings.** We use the context encoder $\phi$ as the pre-trained model for all downstream tasks, following the approach reported by Assran et al. (2023), demonstrating better performance. For representation learning, we globally average pool the output of the feature by the context encoder and use these pooled features as input for the classification head, in alignment with the methodologies described in He et al. (2022a) and Li et al. (2023).

We conducted experiments in both the raw pixel space and the continuous token space. The primary distinction between these two experiments is whether a VAE is utilized to encode the images. For the former, we fully retain the experimental setup of I-JEPA Assran et al. (2023), while for the latter, the training settings are entirely retained from MAGE Li et al. (2023).

To ensure a fair comparison, for D-JEPA-B, we use the checkpoint saved at 600 epochs for the related experiments, and for D-JEPA-L, we use the checkpoint saved at 300 epochs, consistent with the practices in Assran et al. (2023).

| Method | Model | #Params | Acc. | Method | ViT-B | ViT-L |
|---|---|---|---|---|---|---|
| *Gemerative models* | | | | *Representation models* | | |
| BigBiGAN (2019) | RN50 | 23M | 56.6 | data2vec (2022) | - | 77.3 |
| MaskGIT (2022) | BERT | 227M | 57.4 | I-JEPA (2023) | 72.9 | 77.5 |
| ViT-VQGAN (2021) | VIM-Base | 650M | 65.1 | MAE (2022a) | 68.0 | 76.0 |
| ViT-VQGAN (2021) | VIM-Large | 1697M | 73.2 | CAE (2024) | 70.4 | 78.1 |
| D-JEPA (*token space*) | ViT-B | 86M | 46.8 | CMAE (2023b) | 73.9 | - |
| D-JEPA (*token space*) | ViT-L | 304M | 47.6 | MAGE (2023) | 74.7 | 78.9 |
| D-JEPA (*pixel space*) | ViT-B | 86M | 73.1 | MoCo v3 (2021b) | 76.7 | 77.6 |
| D-JEPA (*pixel space*) | ViT-L | 304M | 77.9 | DINO (2021) | 72.8 | - |

Table 9: Top-1 accuracy of linear probing on ImageNet-1K. For reference, we also list the performance of other state-of-the-art representation models as well.

**Linear probing.** Linear probing is a primary evaluation protocol for self-supervised learning. As shown in Table 9, D-JEPA, trained directly on raw pixels, surpasses all generative models in ImageNet-1K linear probe top-1 accuracy, achieving state-of-the-art results. Furthermore, when compared with representation models, D-JEPA remains remarkably competitive.

| Method | ViT-B | ViT-L |
|---|---|---|
| *Patchify on raw pixels* | | |
| Scratch | 82.3 | 82.6 |
| DINO (2021) | +0.5 | - |
| MoCo v3 (2021b) | +0.9 | +1.5 |
| BEiT (2021) | +0.9 | +2.6 |
| MAE (2022a) | +1.3 | +3.3 |
| CAE (2024) | +1.6 | +3.7 |
| MVP (2022) | +2.1 | +3.7 |
| PeCo (2021) | **+2.2** | **+3.9** |
| D-JEPA | +2.0 | +3.2 |
| *Patchify on vector-quantinized tokens* | | |
| Scratch | 80.7 | 80.9 |
| MAGE (2023) | +1.8 | +3.0 |
| MAGE-C (2023) | **+2.2** | +3.4 |
| *Patchify on continuous tokens* | | |
| Scratch | 79.1 | 78.7 |
| D-JEPA | +1.6 | +2.9 |

Table 10: Fine-tuning performance on ImageNet-1K. We report the top-1 accuracy improvement over training from scratch for different methods (other numbers are taken from the respective papers). The ViT models trained from scratch on semantic tokens follow the exact same training settings as the ViT models trained from scratch on original image pixels in He et al. (2022a). It is worth noting that ViT-L's performance is slightly lower than ViT-B's when trained from scratch on continuous tokens, likely due to only 50 epochs of training. However, we used this setting to ensure a fair comparison with other methods.

| Model | cfg | Continuous tokens | | Raw pixels | |
|---|---|---|---|---|---|
| | | FID↓ | IS↑ | FID↓ | IS↑ |
| D-JEPA-B | w/o | 3.40 | 197.1 | 3.45 | 195.2 |
| D-JEPA-B | 3.1 | 1.87 | 282.5 | 1.89 | 281.3 |
| D-JEPA-B | 4.1 | 2.08 | 320.9 | 2.11 | 320.1 |
| D-JEPA-L | w/o | 2.32 | 233.5 | 2.30 | 235.0 |
| D-JEPA-L | 3.0 | 1.58 | 303.1 | 1.61 | 300.7 |
| D-JEPA-L | 3.9 | 1.65 | 327.2 | 1.66 | 330.3 |

Table 11: Comparison of generative performance of D-JEPA trained on continuous tokens and raw pixels.

**Fine-tuning.** Table 10 displays the fine-tuning performance of D-JEPA and other self-supervised learning methods, where all the pre-trained encoder parameters are fine-tuned. It is important to note that directly comparing the final performance of these various methods is inappropriate due to their different data-pacifying approaches. Nonetheless, we can compare their performance improvements relative to training-from-scratch models. The results presented in the tables show that D-JEPA demonstrates strong competitiveness as a representation model. Although its ultimate performance does not surpass that of methods operating in raw space, we observe a substantial performance enhancement compared to training from scratch.

**Analysis on raw pixel space and semantic token space.** Although the D-JEPA model trained in the continuous token space has demonstrated impressive performance on generative tasks, an analysis of the results in Tab. 9 and Tab. 10 reveals that directly training D-JEPA in the pixel space significantly outperforms training in the token space, particularly in linear probing tasks. We attribute this suboptimal performance in both from-scratch and fine-tuning scenarios to using VAE, as discussed in Li et al. (2023). Expressly, the training process of VAEs typically excludes complex image preprocessing techniques such as mixup Zhang et al. (2017b), which impairs the VAE's ability to effectively encode preprocessed images for classification tasks, thereby resulting in a performance deficit. We are optimistic that retraining the VAE with these advanced preprocessing techniques will address this issue and yield improved results.

**Generative performance on raw pixel space and semantic token space.** We evaluated the generative performance of D-JEPA on both the raw pixel space and the semantic token space. As shown in Tab. 11, D-JEPA achieves excellent results in both spaces. Training in the raw pixel space requires significantly more computational resources, which is why we conducted experiments with D-JEPA-B and D-JEPA-L, omitting D-JEPA-H. However, based on the results, we believe that D-JEPA-H can also perform well in the raw space, albeit at the cost of greater computational resources.

## F    FLOW MATCHING LOSS

In this section, we transition from the Gaussian denoising setting as described in Ho et al. (2020) to the flow matching formulation (Ma et al., 2024a; Mo et al., 2024; Lipman et al., 2022), thereby providing additional flexibility to D-JEPA. We adhere to the formulation presented in Zhuo et al. (2024a) and reformulate it for modeling token distribution.

The schedule that defines how to corrupt data to noise significantly impacts the training and sampling of standard diffusion models. Consequently, numerous diffusion schedules, such as VE (Song et al., 2021), VP (Ho et al., 2020), and EDM (Karras et al., 2022), have been carefully designed and utilized. In contrast, flow matching (Lipman et al., 2022; Albergo et al., 2023) emerges as a simple alternative that linearly interpolates between noise and data along a straight line. More specifically, given the data $x_i \sim p(x_i)$ and Gaussian noise $\epsilon \sim \mathcal{N}(0, I)$, we define an interpolation-based forward process:

$$x_i^t = \alpha_t x_i + \beta_t \epsilon,$$

where $\alpha_0 = 0$, $\beta_t = 1$, $\alpha_1 = 1$, and $\beta_1 = 0$. This interpolation for $t \in [0, 1]$ bridges $x_i^0 = \epsilon$ and $x_i^1 = x_i$. Similar to the diffusion schedule, this interpolation schedule offers flexible choices of $\alpha_t$ and $\beta_t$. For example, we can utilize the original diffusion schedules, such as $\alpha_t = \sin\left(\frac{\pi}{2}t\right)$ and $\beta_t = \cos\left(\frac{\pi}{2}t\right)$ for the VP cosine schedule. However, in our framework, we adopt a linear interpolation schedule between noise and data for its simplicity:

$$x_i^t = tx_i + (1 - t)\epsilon.$$

This formulation represents a uniform transformation with constant velocity between the data and noise. The corresponding time-dependent velocity field is defined as:

$$v_t(x_i^t) = \dot{\alpha}_t x_i + \dot{\beta}_t \epsilon \tag{5}$$
$$= x_i - \epsilon, \tag{6}$$

where $\dot{\alpha}$ and $\dot{\beta}$ denote the time derivatives of $\alpha$ and $\beta$. This time-dependent velocity field $v : [0, 1] \times \mathbb{R}^d \to \mathbb{R}^d$ defines an ordinary differential equation known as the Flow ODE:

$$dx_i = v_t(x_i^t)dt.$$

We use $\phi_t(x_i)$ to represent the solution of the Flow ODE with the initial condition $\phi_0(x_i) = x_i$. By solving this Flow ODE from $t = 0$ to $t = 1$, we transform noise into data samples using the approximated velocity fields $v_\theta(x_i^t, t)$. During training, the flow-matching objective directly regresses to the target velocity:

$$\mathcal{L}_v = \int_0^1 \mathbb{E}\left[\| v_\theta(x_i^t, t) - \dot{\alpha}_t x_i - \dot{\beta}_t \epsilon \|^2\right] dt, \tag{7}$$

which is termed the Conditional Flow Matching loss (Lipman et al., 2022), sharing similarities with the noise prediction or score prediction losses in diffusion models.

**Analysis of flow matching loss**    We experimented to evaluate the performance of flow matching loss using the D-JEPA-B model on the ImageNet1k dataset for image generation, training the model for 300 epochs. The experimental settings were consistent with those described in Sec. 4. The FID convergence curve is illustrated in Fig. 9. As anticipated, the flow matching loss demonstrated a faster convergence rate than the diffusion loss, achieving convergence within approximately 160 epochs.

However, it was observed that the flow matching loss encounters difficulties in reducing the FID score during the later stages of training. We hypothesize that this limitation may be attributed to the denoising MLP being too lightweight to manage the continuous-space denoising operations required

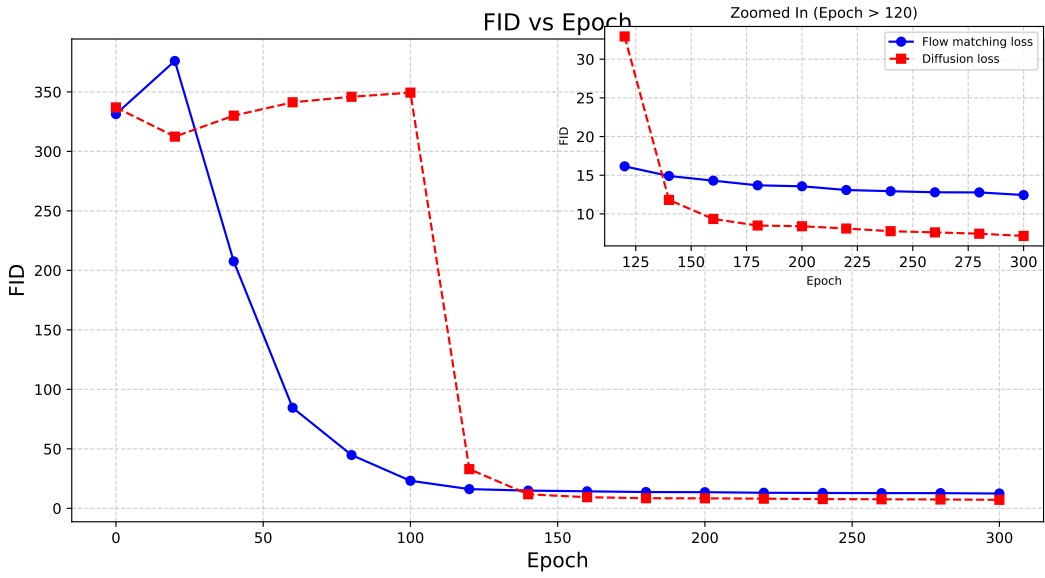

Figure 9: Comparison between flow matching loss in Eq. 7 and diffusion loss in Eq. 4 on FID.

by flow matching efficiently. This issue can potentially be alleviated by increasing the network capacity of the denoising MLP. Despite this, the images generated using flow matching loss exhibit superior texture quality.

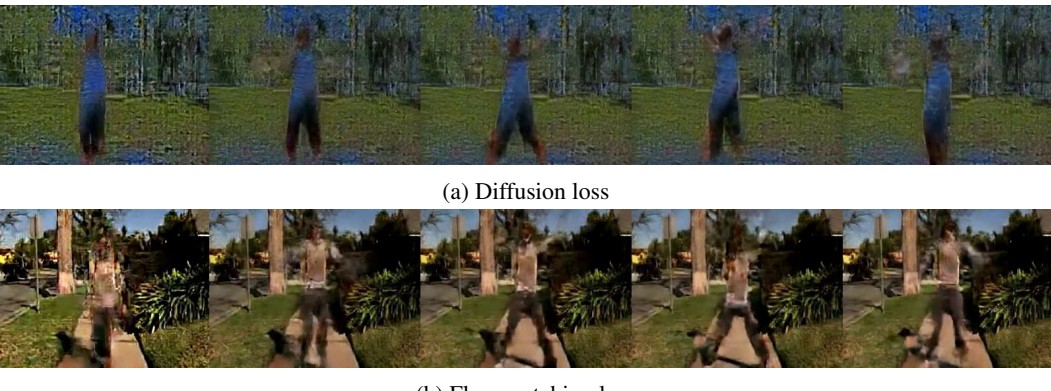

Figure 10: Failure cases of diffusion loss in video clip generation. The samples are generated using the D-JEPA-L model trained for 1600 epochs on UCF101 dataset, with 225 auto-regressive steps for sampling 3000 tokens.

As illustrated in App. G, the flow matching loss significantly outperforms the diffusion loss for more challenging tasks. In specific scenarios, the diffusion loss may even lead to training failures, as depicted in Fig. 10. Consequently, we adopt the flow-matching loss for the remainder of this paper.

## G  COMPREHENSIVE EXPERIMENTS

This section presents detailed experimental procedures and results for D-JEPA across various tasks. It is important to emphasize that *our objective is not to achieve state-of-the-art performance on these tasks, as this is beyond the scope of the current study and will be explored in future research.* Instead, we aim to demonstrate that D-JEPA is a robust and versatile generative model capable of handling various continuous data types. Consequently, we did not perform extensive hyperparameter tuning;

unless otherwise specified, all training settings are consistent with those used for image generation on ImageNet. As noted in Appendix F, we employ flow matching loss instead of diffusion loss for all experiments in this section.

## G.1 TEXT-TO-AUDIO GENERATION

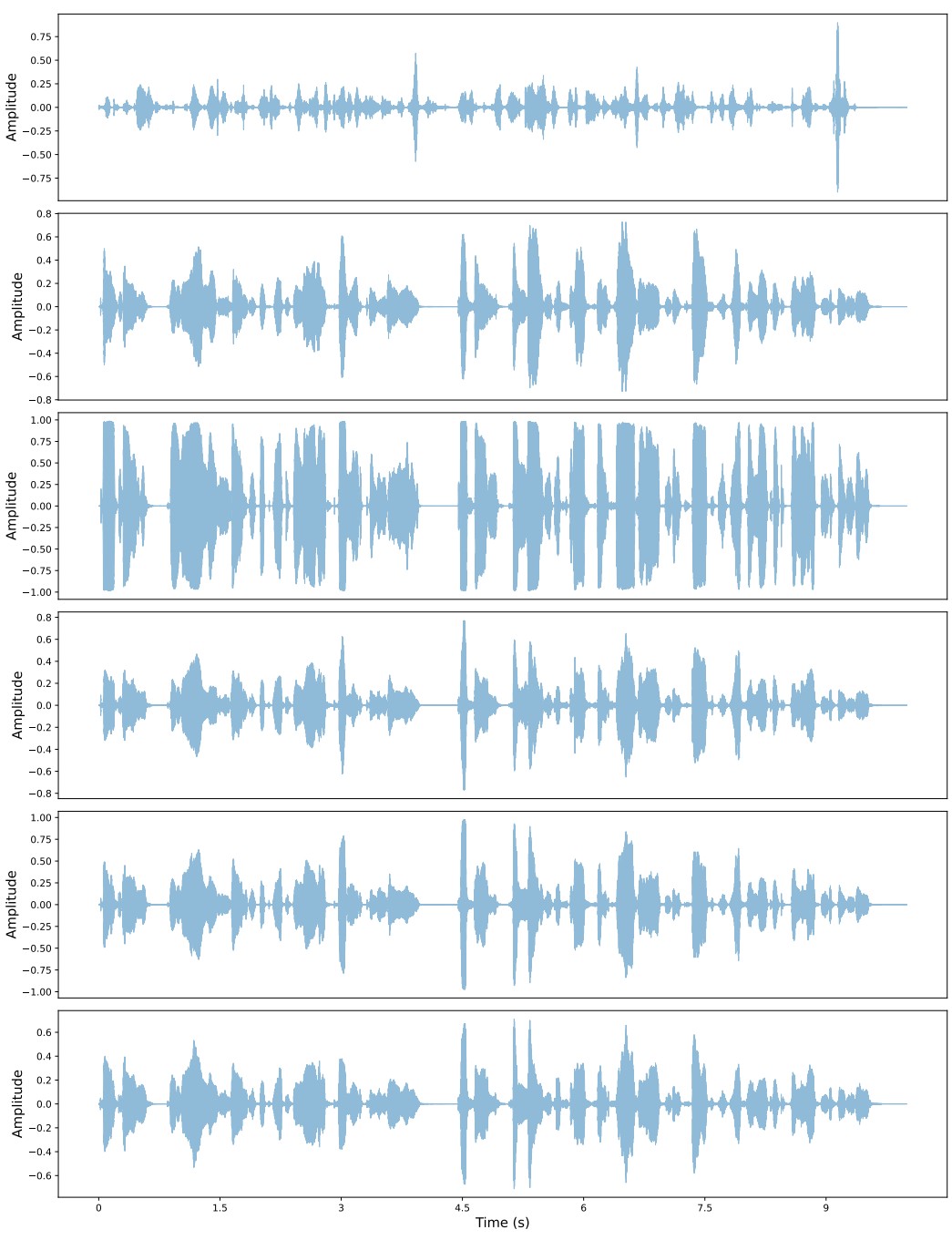

Figure 11: Visualizations of the generated waveform over training epochs. From top to bottom, each waveform represents the result sampled every 400 epochs. The corresponding text of the generated speech samples is "*Printing, in the only sense with which we are at present concerned, differs from most if not from all the arts and crafts represented in the Exhibition.*"

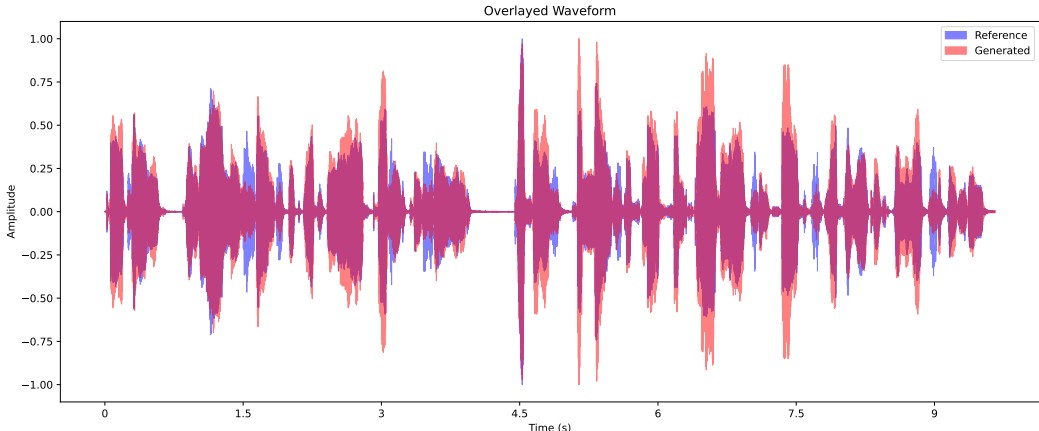

Figure 12: Visualizations of the reference and generated waveform. The corresponding texts of generated speech samples is "*Printing, in the only sense with which we are at present concerned, differs from most if not from all the arts and crafts represented in the Exhibition*".

In this experiment, we adapted D-JEPA-B for speech synthesis, incorporating a phoneme and pitch encoder. To integrate text conditions, we replaced the attention module in the vision transformer with multimodal attention, as first proposed by Esser et al. (2024). The phoneme vocabulary size is set to 73. For the pitch encoder, the lookup table size is set to 300, and the encoded pitch embedding size is 256, with a hidden channel size of 256.

We utilized the LJSpeech benchmark dataset (Ito, 2017), which consists of 13,100 audio clips sampled at 22,050 Hz from a female speaker, totaling approximately 24 hours. Text sequences were converted into phoneme sequences using an open-source grapheme-to-phoneme conversion tool (Sun et al., 2019). Following common practices (Chen et al., 2021a; Min et al., 2021; Zhuo et al., 2024a), we preprocessed the speech and text data by (1) extracting the spectrogram with an FFT size of 1024, hop size of 256, and window size of 1024 samples; (2) converting it to a mel-spectrogram with 80 frequency bins; and (3) extracting the F0 (fundamental frequency) from the raw waveform using Parselmouth. Our implementation is based on the open-source project by Huang et al. (2023a).

Notably, the diffusion loss did not perform optimally on this small-scale dataset. To better accommodate the varying lengths of sentences, we employed 1D Rotary Position Embedding (RoPE) (Su et al., 2024) as the positional embedding. D-JEPA-B was trained for 134,000 steps, approximately 2,600 epochs, using 4 NVIDIA A100 GPUs with a total batch size of 256 sentences. HiFi-GAN (Kong et al., 2020) (V1) was utilized as the vocoder to synthesize the waveform from the generated mel-spectrogram.

**Analysis** In Fig. 11, we demonstrate the sampling results of the trained model at intervals of 400 epochs. We observed that after 1,600 epochs, D-JEPA-B can generate relatively clear audio. As training proceeded, the speech quality steadily improved, with pronunciations becoming clearer and the intonation and pauses closely matching those of natural speech. Fig. 13 and Fig. 12 show sampled mel-spectrogram and waveform, respectively. It is evident that after 2,600 epochs of training, the sampled audio closely resembles the reference audio.

### G.2 TEXT-TO-IMAGE GENERATION

In this experiment, we adapted D-JEPA-L for the task of text-to-image generation. Similar to previous experiments, we incorporated text conditions using multimodal attention. Additionally, we utilized a more powerful VAE model from Esser et al. (2024). A $256 \times 256$ resolution image is encoded into a latent representation of size $16 \times 32 \times 32$, which is then patched with a patch size of 2. We replaced the positional encoding with a 2D RoPE suitable for images for this task. We employed the QWen2-1.5B language model (Yang et al., 2024a) as the text encoder, which offers superior text

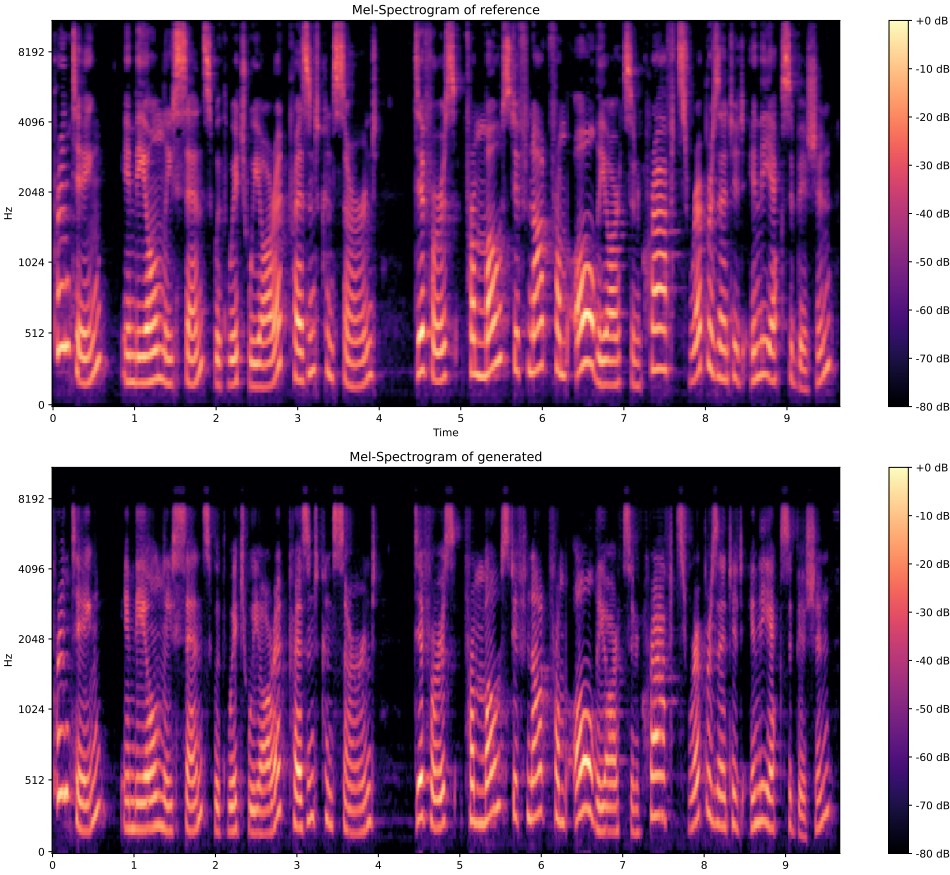

Figure 13: Visualizations of the reference and generated mel-spectrograms. The corresponding texts of generated speech samples is "*Printing, in the only sense with which we are at present concerned, differs from most if not from all the arts and crafts represented in the Exhibition*".

understanding capabilities compared to models like CLIP (Radford et al., 2021) or T5 (Raffel et al., 2020).

We conducted experiments on an internal dataset composed of both public and private data. The public dataset comprises millions of images from Sun et al. (2024). We performed meticulous data cleaning, referencing the methods used by PixelArt (Chen et al., 2023) and SD3.0 (Esser et al., 2024). Since we did not intend to explore multi-scale resolution generation at this stage (reserved for future work), we resized all images to a $256 \times 256$ resolution for training.

We trained D-JEPA-L from scratch for approximately 500 epochs (measured by the ImageNet data volume) without additional pre-training. Unlike many diffusion models such as Chen et al. (2023), we found that training from scratch yielded satisfactory results for D-JEPA-L; using pre-trained weights from ImageNet resulted in decreased performance, consistent with the findings in Zhuo et al. (2024a). We conducted this experiment on four workers, each with 8 H800 GPUs. We maintained a total batch size of 2048 with the help of gradient checkpointing techniques.

**Analysis** In Fig. 14, we show the images sampled every 40 epochs during the early training stages (up to 200 epochs). Early in training, D-JEPA tended to generate single objects and struggled to understand multiple objects or spatial relationships. After 120 epochs of training, the model could generally generate single objects well, but various objects were still missing. As training progressed, D-JEPA began generating multiple objects and gradually grasped the correct spatial relationships. We observed that after about 160 epochs, the model could generate high-quality images, as shown in the penultimate image in Fig. 14. However, it still struggled to create accurate spatial relationships

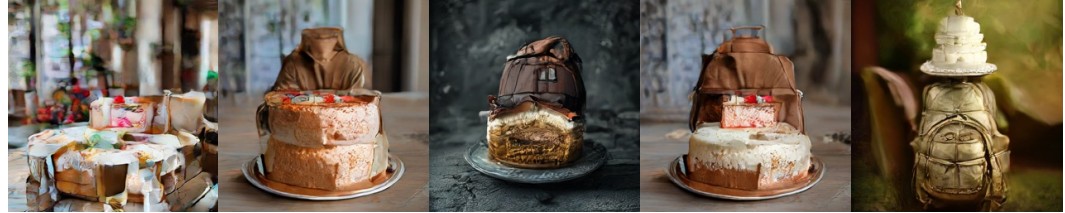

Figure 14: Text to image samples during training. From left to right, the results are sampled at intervals of 40 epochs, from the 40th to the 200th epoch. The corresponding prompt: *a photo of a backpack below a cake.*

between objects. For instance, when given the prompt "a backpack **below** a cake," the generated content showed "a backpack **behind** a cake," which is more consistent with common sense. In other words, the model could understand conventional spatial relationships at this stage but still struggled with abstract spatial relationships, especially those not present in the training data. As training continued, the model's ability to understand abstract concepts improved, as demonstrated in the last image in the figure, which correctly generated the spatial relationship.

In Fig. 15, we demonstrate the model's capabilities to generate high-quality images after *300* training epochs. D-JEPA can already produce detailed portraits and rich scenes, although some local artifacts still exist. We believe that by using larger models, conducting more extensive training, and utilizing higher-resolution data, D-JEPA's text-to-image generation capabilities will be significantly enhanced.

## G.3 CLASS CONDITIONED VIDEO GENERATION

In this experiment, we adapted D-JEPA-L for class-conditioned video generation. We utilized the open-source VideoVAE from Yang et al. (2024b), which compresses the video's temporal and spatial resolutions by factors of 4 and 8, significantly reducing the computational burden during video generation. We divided each video frame into tokens with a patch size of 2 along the spatial and four along the temporal dimensions. Empirically, we found that using a larger patch size in the temporal dimension helps the model learn more consistent motion.

We conducted the experiments on the UCF101 dataset (Soomro, 2012), a widely recognized benchmark in human action recognition, consisting of 13,320 video clips spanning 101 action categories, including a diverse range of activities such as sports, musical instruments, and human-object interactions. This richly annotated dataset covers a broad spectrum of real-world scenarios captured under varying conditions, making it an essential resource for advancing and evaluating video generation methods. It has been widely used in recent works (Ma et al., 2024b; Ho et al., 2022). We trained our model on the original $240 \times 320$ resolution. We sampled eight frames per second to generate 20-second videos, totaling 160 frames. We used zero padding for videos shorter than 20 seconds to align the number of frames. During training, each input video clip was sized at $160 \times 240 \times 320$, yielding a latent embedding with $40 \times 30 \times 40$, and subsequently patched into $10 \times 15 \times 20 = 3000$ tokens. During sampling, we completed video generation in 64 steps. We trained D-JEPA-L from scratch on four workers, each equipped with 8 H800 GPUs, maintaining a global batch size of 512. We also utilized gradient checkpointing to reduce the demand for GPU memory.

| | DIGAN (2022) | StyleGAN-V (2022) | Latte* (2024b) | LVDM (2022b) | MoStGAN-V (2023) | PVDM (2023c) | D-JEPA |
|---|---|---|---|---|---|---|---|
| FVD-16 | 1630.2 | 1431.0 | 333.61 | 372.0 | 1380.3 | 1141.9 | 365.2 |

Table 12: Quantitative comparisons of short video generation on UCF101. * indicates training on additional image data.

**Analysis**   Current video generation models based on diffusion typically require pre-training or joint training on images and videos. We found this approach unnecessary for video generation; instead, it introduces an artificial separation between spatial and temporal aspects, hindering motion generation. The D-JEPA-based video generation model is trained directly on videos, treating all tokens as standard tokens without incorporating 3D attention or other structural designs. These

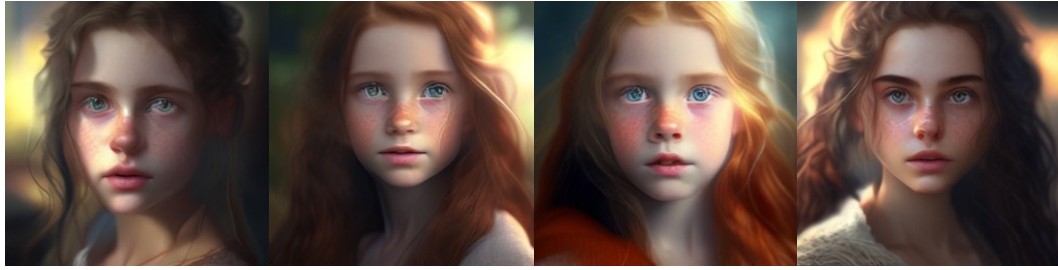

(a) A beautiful girl.

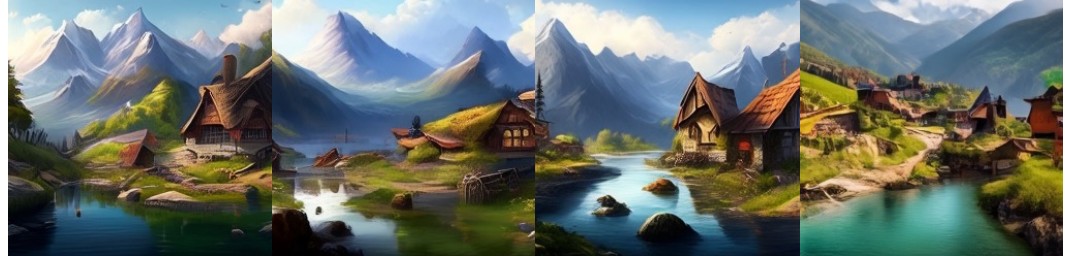

(b) Beautiful scene with mountains and rivers in a small village.

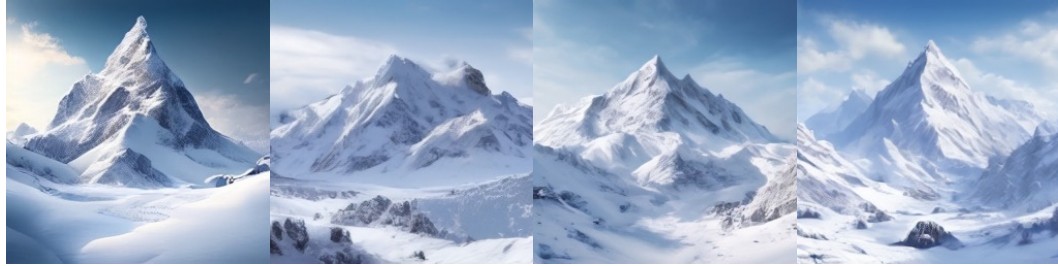

(c) A snowy mountain.

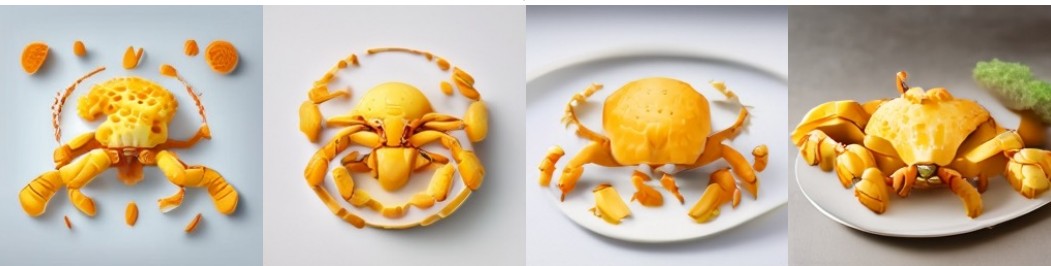

(d) A crab made of cheese on a plate.

Figure 15: Samples of text to image generation. Resolution: $256 \times 256$.

straightforward yet effective mechanisms enable D-JEPA to generate videos more holistically. Since D-JEPA randomly generates the next token from the entire token sequence, it captures the overall video quality more effectively. It decides when to terminate video generation, thereby avoiding the production of redundant video segments, as illustrated in Fig. 10. Fig. 16 also indicates that D-JEPA maintains stability throughout the training process. We also present the FVD metrics in Table 12, where D-JEPA achieves results second only to Latte (Ma et al., 2024b), which is trained on additional image data.

### G.4  MULTI-MODAL GENERATION

In this experiment, we aim to demonstrate the potential of D-JEPA as a unified multimodal model. This is an ambitious attempt, as D-JEPA can generate text using *next token prediction* and create im-

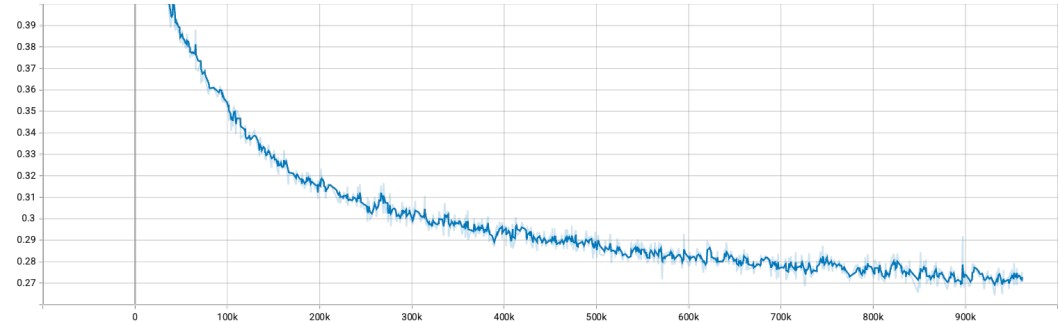

Figure 16: The training curve of flow matching loss on UCF101.

ages, videos, etc., using *next set of tokens prediction*. Essentially, we adopted the design philosophy of Transfusion (Zhou et al., 2024) but replaced the diffusion model with the D-JEPA model.

To simplify the task, we constructed a model that simultaneously generates text and images. Unlike the text-to-image generation described in App. G.2, image generation in the multimodal mode does not rely on any text embedding model. Text and images are directly used as inputs to the model, which then simultaneously predicts both text and image.

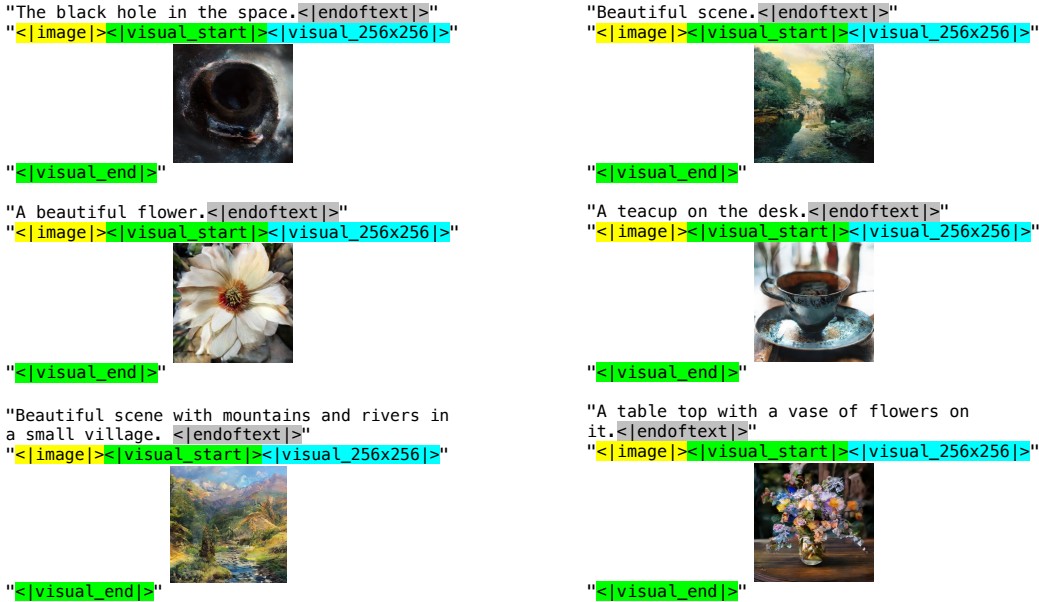

Figure 17: Samples of text to image generation by multi-modal models build on the top of D-JEPA.

During the sampling process, we input the image caption that needs to be generated and let the network continuously predict the next tokens until it samples a "$< |visual\_start| >$" control token. Then, it switches to the next set of tokens prediction mode to complete the sampling of the entire image. The specific sampling process and training details follow the methodology described by Zhou et al. (2024). We showcase the image generation effects of this novel architecture in Fig. 17.

Empirically, we found that the multimodal model built with D-JEPA can more effectively couple discrete text and other continuous data types in terms of architecture and engineering implementation. Further exploration of this field will be presented in future work.

## H  LIMITATIONS

**Potential performance bottleneck from denoising MLP.**  In the D-JEPA framework, the context encoder, target encoder, and feature predictor all utilize standard visual transformer structures. This work and numerous other studies extensively validated the scalability and capacity for scaling up these structures, suggesting significant potential for D-JEPA. However, diffusion or flow matching loss is implemented using a simple Multi-Layer Perceptron (MLP) structure. Although it is theoretically proven that a 3-layer MLP can fit any function, we have observed in practice that the MLP may become a bottleneck for the final performance of the D-JEPA model. More critically, our experiments indicate that increasing the number of MLP layers or widening the MLP dimensions does not significantly improve performance. Recent works, such as KAN (Liu et al., 2024), have demonstrated significant advantages over MLPs in various tasks. We also attempted to construct a network of comparable scale using KAN, but so far, it has not resulted in a significant performance improvement. While the current MLP structure has achieved satisfactory results in our experiments, exploring alternative network structures to replace the MLP remains a crucial area for further investigation.

**Efficiency issues due to bi-directional attention.**  Although D-JEPA employs a generalized next-token prediction strategy, its primary structure is entirely realized through visual transformers, which use bi-directional attention during training and inference. This prevents highly effective key-value caching strategy based on causal attention to reduce computation. Notably, in the feature predictor network, we train with all masked and context tokens together using bi-directional attention. This means that each token must continuously attend to information from context and masked tokens. This behavior during training forces us to feed all tokens into the network simultaneously during sampling, especially at the beginning, leading to substantial unnecessary computational overhead. Simple estimates suggest that this process can result in up to $50\%$ useless computation for the feature predictor in extreme cases.

Intuitively, when predicting the features of masked tokens, we should only need information from context tokens without attending to other masked tokens. This would allow us to avoid feeding all masked tokens into the network during inference, significantly reducing extra computational costs. Based on this, we attempted to develop a new attention mechanism during training. In this approach, the attention for masked tokens in the feature predictor is restricted to all context tokens and itself, while context tokens always attend to all context tokens.

Surprisingly, training with this new attention mechanism did not effectively reduce the FID on ImageNet (FID remained above 50 after 160 epochs). This aligns with findings in Zhou et al. (2024), emphasizing that diffusion models must use bi-directional attention during training to ensure effectiveness. Similar observations were made in the MAR (Li et al., 2024) context. Therefore, improving the efficiency of the attention module during training and inference for D-JEPA remains a critical issue.

