# OpenReview forum: "Denoising with a Joint-Embedding Predictive Architecture"
_ICLR.cc/2025/Conference — ICLR 2025 Poster_

### Official Review · Reviewer_5MfA · 2024-10-26

**Soundness:** 2
**Presentation:** 3
**Contribution:** 2
**Rating:** 6
**Confidence:** 4

**Summary:**

The paper, D-JEPA, explores a novel approach by merging joint-embedding predictive architectures with generative modeling. It applies masked image modeling as a generalized next-token prediction strategy for autoregressive data generation. D-JEPA employs a combination of diffusion losses and MAE loss to model token distributions effectively. It aims to enhancing generative tasks across various domains like images, videos, and audio. Extensive experiments demonstrate the model's scalability and effectiveness, claiming state-of-the-art performance on ImageNet class-conditional benchmarks.

**Strengths:**

The ideas tha integrating joint-embedding predictive architectures(e.g. MAE and MIM) with diffusion-based generative modeling makes sense.

The experiments in the paper are robust, covering a wide range of applications, which showcases the versatility of the D-JEPA model. This extensive experimental validation strengthens the paper's claims about the model's effectiveness across different types of generative tasks.

**Weaknesses:**

However, the paper presents a significant methodological oversight—it appears as a mere combination of existing MAE (Masked Autoencoder) and MAR (Masked Autoregressive Model) methods without a critical analysis or ablation study showing how the MAE branch impacts the AR branch. Specifically, the absence of experiments where the MAE branch is removed casts doubt on the incremental benefit of this integration.

The representation learning capability of D-JEPA is also questionable. According to Table 7, the best ImageNet accuracy is obtained when the model is trained on pixel-level data, while the generative experiments focus on a D-JEPA trained on VAE latent space. This discrepancy points to a lack of a truly unified model approach, as the performance seems to depend heavily on the training specifics rather than the model architecture itself.

**Questions:**

1. Could the authors provide an ablation study that isolates the MAE branch to elucidate its specific impact on the AR branch's performance?

2. How does the model's performance on representation learning tasks vary between training in pixel space and latent space, and what does this indicate about the model's ability to serve as a unified architecture?

3. It’s interesting to see from Tables 1 and 2 that D-JEPA has more parameters than MAR. I’m curious—shouldn’t these models have similar architectures? Can the authors explain why D-JEPA is larger? What additional components does D-JEPA include that MAR doesn't?

---

> ### Author Response · Authors · 2024-11-18
> **Replying to Q1**
>
> ## However, the paper presents a significant methodological oversight—it appears as a mere combination of existing MAE (Masked Autoencoder) and MAR (Masked Autoregressive Model) methods without a critical analysis or ablation study showing how the MAE branch impacts the AR branch. Specifically, the absence of experiments where the MAE branch is removed casts doubt on the incremental benefit of this integration.
>
> D-JEPA is not based on the MAE structure; it is an improvement upon the JEPA structure, both utilizing the same ViT structure as the backbone. The effectiveness and design details of JEPA have been well-validated in existing literature, such as in "I-JEPA: Self-Supervised Learning from Images with a Joint-Embedding Predictive Architecture." The primary improvement in D-JEPA is the integration of the generative capabilities of diffusion models into JEPA, enabling it to handle generative tasks.

---

> ### Author Response · Authors · 2024-11-18
> **Replying to Q2**
>
> ## The representation learning capability of D-JEPA is also questionable. According to Table 7, the best ImageNet accuracy is obtained when the model is trained on pixel-level data, while the generative experiments focus on a D-JEPA trained on VAE latent space. This discrepancy points to a lack of a truly unified model approach, as the performance seems to depend heavily on the training specifics rather than the model architecture itself.
>
> For discussions on the representation learning capability of D-JEPA, please refer to the analysis provided [here](https://openreview.net/forum?id=d4njmzM7jf&noteId=IvlD33x5eq). As analyzed in Appendix E, the suboptimal representation learning capability of D-JEPA in latent space is primarily due to the VAE's inability to effectively encode images after data augmentation. This results in D-JEPA performing better in pixel space than in latent space, and our experimental results support this hypothesis. Additionally, similar analyses were conducted in MAGE.

---

> ### Author Response · Authors · 2024-11-18
> **Replying to Q3**
>
> ## Could the authors provide an ablation study that isolates the MAE branch to elucidate its specific impact on the AR branch's performance?
>
> For an ablation study that isolates the impact of the JEPA branch on generative performance, please refer to the discussion [here](https://openreview.net/forum?id=d4njmzM7jf&noteId=lS8O0LbtPH). In summary, the JEPA branch in D-JEPA effectively enhances the model's convergence speed.

---

> ### Author Response · Authors · 2024-11-18
> **Replying to Q4**
>
> ## How does the model's performance on representation learning tasks vary between training in pixel space and latent space, and what does this indicate about the model's ability to serve as a unified architecture?
>
> D-JEPA performs better in pixel space than in latent space, likely due to the VAE's limitations in effectively encoding images after data augmentation. Currently, the core focus of this work is to address generative capabilities in multimodal models, while representation learning, which leans more towards understanding, is not within the scope of this work. However, our recent work has validated D-JEPA's effectiveness in constructing unified multimodal generative and understanding models, where multimodal joint training can significantly overcome these limitations.

---

> ### Author Response · Authors · 2024-11-18
> **Replying to Q5**
>
> ## It’s interesting to see from Tables 1 and 2 that D-JEPA has more parameters than MAR. I’m curious—shouldn’t these models have similar architectures? Can the authors explain why D-JEPA is larger? What additional components does D-JEPA include that MAR doesn't?
>
> We provided a detailed explanation [here](https://openreview.net/forum?id=d4njmzM7jf&noteId=Vh2BYSKA09). In summary, D-JEPA adopts the standard ViT B/L/H structure, whereas MAR uses a non-standard ViT B/L/H structure, leading to discrepancies in model parameters between the two.

---

> ### Comment · Reviewer_5MfA · 2024-11-22
>
> Thank you to the authors for the clarification. I now have a better understanding of the method and architecture. However, I still have some concerns:
>
> **Ablation Study**
>
> I could not find the experiments I previously requested, where the **I-JEPA branch is removed and MAR branch is evaluated in isolation**, keeping the other design aspects unchanged. The link provided by the authors directs to the current page, which does not include experiments related to this specific ablation.
>
>
> **Representation learning**
>
> In the rebuttal, the authors state:
>
> > Currently, the core focus of this work is to address generative capabilities in multimodal models, while representation learning, which leans more towards understanding, is not within the scope of this work.
>
> However, this claim contradicts the content of the paper. The authors explicitly position their work as aiming to **unify multimodal models** and emphasize balancing `representation learning and diffusion learning`. For example, on `Line 980`, the paper states:
>
> > Our work uniquely verifies the mutual benefits of representation learning and generative modeling
>
> This demonstrates that representation learning is indeed within the scope of the work, as evidenced by the extensive experiments in the appendix. The rebuttal appears inconsistent with the paper’s claims.

---

> > ### Author Response · Authors · 2024-11-23
> >
> > Thank you for your continued interest in our work.
> >
> > I apologize for the inconvenience caused by the links in my previous response not directing correctly to the intended content. In my response to Reviewer gTwo's Q2, I explained the details regarding the ablation study. I suspect that the issue might be due to the webpage configuration of OpenReviewer, which resulted in the hyperlinks not functioning correctly. I apologize for any confusion and the time this may have taken. In the revision version, we have reorganized the content of the ablation study, which you can now find in Appendix C.3.
> >
> > For your convenience, here is an excerpt from my response to Reviewer gTwo's Q2:
> >
> > ```
> > ## There is no comparison to baseline methods, such as the effect of removing the JEPA loss.
> >
> > There is indeed an ablation study in the paper, although not explicitly compared and analyzed. Here we provide the corresponding analysis. We conducted ablation experiments on the D-JEPA-B model. The results of MAR-B serve as the baseline (i.e., the results without the JEPA Loss), because MAR-B and D-JEPA-B have identical model structures and experimental parameters, making them a comparable baseline.
> >
> > From Fig. 6, we can see that D-JEPA-B achieves the performance of MAR-B at 600 epochs. Additionally, from Tab.1 and Tab.2, we observe that D-JEPA-B significantly outperforms MAR-B at 1400 epochs. For a more intuitive comparison, we provide some key data points below:
> >
> > |  Model   | Epoch |  FID  |
> > | :------: | :---: | :---: |
> > |  MAR-B   |  800  | 3.48  |
> > | D-JEPA-B |  600  | 3.50  |
> > | D-JEPA-B |  800  | 3.43  |
> > | D-JEPA-B | 1400  | 3.40  |
> >
> > These results are obtained without using classifier-free guidance, which we believe better reflects the true scenario.
> > ```
> >
> > I hope this information helps you better understand our work. If you have any further questions or need more clarification, please feel free to reach out. Thank you again for your patience and understanding.

---

> > > ### Comment · Reviewer_5MfA · 2024-11-23
> > >
> > > Thank you for providing the additional explanation.
> > >
> > > **Ablation Study**
> > >
> > > As clarified in the rebuttal:
> > >
> > > >  D-JEPA adopts the standard ViT B/L/H structure, whereas MAR uses a non-standard ViT B/L/H structure
> > >
> > > This indicates that comparing `MAR-B` and `D-JEPA-B` alone does not result in a completely fair ablation study. A proper ablation study requires all hyperparameters and architectures to be identical.
> > >
> > > **Representation learning**
> > >
> > > While I appreciate the revision, I believe the confusion stems from more than just one sentence. The introduction broadly discusses the interaction between representation learning and generation, which could mislead readers.
> > >
> > > I strongly recommend revising the introduction to clearly emphasize that the `the core of current study is the generation, rather than a unified model`. Additionally, any claims in the paper regarding a `unified multimodal model` should be removed to avoid further misunderstanding.

---

> > > > ### Author Response · Authors · 2024-11-23
> > > >
> > > > Thank you for your prompt feedback!
> > > >
> > > > **Regarding the ablation study**, we have thoroughly verified that the network structures of D-JEPA-B and MAR-B are identical, both based on ViT-B. However, I'm not sure why MAR-L and MAR-H did not adopt the standard ViT-L and ViT-H structures. Therefore, the comparison in the ablation study between D-JEPA-B and MAR-B is fair. In our actual code implementation, we indeed based it directly on the open-source code of MAR, with the only addition to D-JEPA-B being an MLP layer introduced by the JEPA loss, which adds about 4 million parameters.
> > > >
> > > > **Concerning the description** in the paper related to the unified multimodal model, it indeed seems somewhat inappropriate. Although we showcased generative results across various modalities in Appendix G, they primarily focus on continuous modality generation experiments and have not fully validated the model's understanding capabilities. Given this, we will carefully revise the relevant descriptions in the paper soon. A more appropriate description might be "unified multimodal generation model," which would be more accurate.
> > > >
> > > > Thank you again for actively participating in the discussion, and we hope you will look forward to the upcoming revised version.

---

> > > > ### Author Response · Authors · 2024-11-23
> > > > **Thank you very much for actively participating in the rebuttal process and providing valuable suggestions for revisions.**
> > > >
> > > > Thank you very much for actively participating in the rebuttal process and providing valuable suggestions for revisions. In the latest revision of the paper, we have carefully revised sections that may have been unclear, particularly in the introduction and subsequent experimental analysis sections. We hope that these modifications will further reduce any confusion regarding the paper.

---

> > > > > ### Comment · Reviewer_5MfA · 2024-11-23
> > > > >
> > > > > Thank the author for the explanation. Some of my concerns have been addressed, so I am increasing my score to 6. Best of luck!

---

> > ### Author Response · Authors · 2024-11-23
> >
> > Thank you very much for pointing out the inappropriate expressions in Appendix A regarding related works. We indeed made an incorrect statement around line 980 concerning D-JEPA's contributions to representation learning, which led to misunderstandings. In the summary of contributions at the end of the Introduction, we did not list D-JEPA's performance in representation learning as one of the contributions of this paper. The main text of the paper also highlights its strong performance in generative tasks. Therefore, we have corrected the erroneous statement around line 980 in the revision version.
> >
> > Here is the version before the correction:
> >
> > Recent advances such as MAGE~\citep{li2023mage} achieve high-quality representations and image generation. However, these methods are limited to image data, *failing to empirically demonstrate mutual reinforcement between tasks. Our work uniquely verifies the mutual benefits of representation learning and generative modeling through empirical evidence, demonstrating how these processes can be synergistically combined for superior performance across different types of continuous data.*
> >
> > Here is the revised version:
> >
> > Recent advances such as MAGE~\citep{li2023mage} achieve high-quality representations and image generation. However, these methods are limited to image data, **and have not been validated on other types of continuous data such as speech and video. Our work uniquely demonstrates the significant impact of representation learning across image, audio, and video generation, highlighting its versatility and effectiveness in these domains.**
> >
> > Thank you again for your valuable feedback on this work. Additionally, we have included more experimental results on representation learning in the revised version.

---

### Official Review · Reviewer_3tNA · 2024-10-29

**Soundness:** 3
**Presentation:** 2
**Contribution:** 3
**Rating:** 8
**Confidence:** 3

**Summary:**

The author describes a JEPA based architecture which combines a prediction loss and a diffusion loss.
* The prediction loss learns to predict the representation of the mask tokens
* The diffusion loss learns to predict the raw pixels (or the latent representation of the patch in case of VAE)

The architecture itself is based on a three ViTs:
* The context encoder takes the unmasked tokens and outputs Y
* The feature predictor takes Y, pad it, and learns to reconstruct the full representation Z
* The target encoder takes all tokens (no masking) and produces the target representation Z'
* The prediction loss is a smooth l1 loss between Z and Z'
* The diffusion loss takes Z as input and tries to predict the pixels / latents
* The target encoder is itself an EMA of the context encoder

The paper explores this architecture and shows that:
1) The diffusion loss prevents the collapse of traditional JEPA architectures
2) This architecture is scalable, as the results improves with increased compute budget
3) At inference, instead of predicting all unmasked tokens, we can predict a subset of them, as do MaskGIT, to provide better results
4) They reach SOTA performance on class conditional generation on ImageNet 256x256, equating the results of MAR with CFG

**Strengths:**

* Interesting combination of JEPA representation learning with generative AI, showing that representation learning can help generative AI
* Strong SOTA results on ImageNet 256x256, better or equal than MAR
* When applied for representation learning, the context encoder achieves good results on ImageNet classification
* Experiments showing generalization to text conditioned image generation (and not just class conditioned)
* Experiments showing that it works with audio as well, class conditioned video generation, and CFG

**Weaknesses:**

* It would be interesting to have generation results at higher resolution than 256x256 px to see if inference speed suffers when bi-directional attention meets lots of tokens

**Questions:**

n/a

---

> ### Author Response · Authors · 2024-11-18
> **Replying to Q1**
>
> ## It would be interesting to have generation results at higher resolution than 256x256 px to see if inference speed suffers when bi-directional attention meets lots of tokens.
>
> We have showcased the generation results of 1024x1024 images [here](https://openreview.net/forum?id=d4njmzM7jf&noteId=houkfsPAB9) or you can click [here](https://anonymous.4open.science/r/DJEPA-T2I-Examples-389C/README.md). Some quantitative metrics are presented [here](https://openreview.net/forum?id=d4njmzM7jf&noteId=WMuPFBJ94j). During our experiments, we observed that bi-directional attention does indeed lead to a significant decrease in inference speed when generating a large number of tokens. However, we can mitigate this issue by using a next set-of-tokens prediction approach, which allows us to generate multiple tokens at once, thereby greatly reducing the number of inference steps needed for image generation. In our given samples, we were able to generate high-quality 1024x1024 images using only 128 autoregressive steps.

---

> > ### Comment · Reviewer_3tNA · 2024-11-18
> > **1024x1024 Generations**
> >
> > Thank you so much for the 1024px generations! Out of curiosity, how much time does it takes to do 128 AR steps at 1024px resolution? Also thank you for the 512px FID results listed above, it is interesting to see that you get SOTA results, but that CFG does not seem to help as much as for other method. Something interesting to investigate I guess.

---

> > > ### Author Response · Authors · 2024-11-19
> > >
> > > Thank you for your interest in our work.
> > >
> > > **Regarding inference speed for generating 1k images**: To be transparent, without employing any special acceleration strategies (such as Triton deployment), our naive PyTorch code takes about 70 seconds on average to generate four 1k images on an A100-80GB GPU, consuming around 40GB of memory. This is with the D-JEPA model size of 2.6B, using the CFG strategy, and 128 autoregressive steps. If the CFG strategy is not used, the speed doubles, and image generation speed is inversely proportional to the number of autoregressive steps. Notably, we found that 128 steps typically yield more stable results, performing better than even 256 or 512 steps, a phenomenon consistent with other observations in Imagenet experiments. To further improve efficiency, experimenting with causal attention seems intuitive. However, we found that causal attention significantly degrades overall performance (FID around 50). Similar observations have been made in recent works like Transfusion, EMU3, and Show-o. Aggregating similar features, such as through pixel-art sigma techniques, might effectively address the sampling efficiency challenge for high-resolution images.
> > >
> > > **Regarding the impact of CFG on D-JEPA**: CFG dramatically affects FID. During early training stages, we noticed that without CFG, FID can exceed 7, while with CFG, FID may directly drop to between 3 and 5. This phenomenon is reflected in Tab.1 and Tab.2. Our analysis in Appendix D suggests that metrics without CFG might better reflect actual model performance. This is because the impact of CFG on FID varies by model. As for 512 resolution, CFG also significantly affects FID, provided the FID score isn't already very low. When the FID without CFG already approaches the theoretical minimum (around 1.x), the improvements brought by CFG become minimal.

---

### Official Review · Reviewer_dQp1 · 2024-11-02

**Soundness:** 2
**Presentation:** 1
**Contribution:** 2
**Rating:** 5
**Confidence:** 5

**Summary:**

The proposed D-JEPA is the combination of Joint-embedding Predictive Architecture (I-JEPA) with MAR's dfiffusion parts, ie, leveraging a feautre from JEPA as the condition of diffusion parts.  Good generative performance is achieved by the proposed model.
And authors provide the exps over multi-modalities.

**Strengths:**

1. Suffieicnt experiments, including generative reuslts on Imagenet-256, sufficient abaltion studies and the exps about representation learning.

2. Authors present the effectiveness of D-JEPA  over multi-modalities including videos, images and audio.

3. Authors provide a theretical support about the proposed models.

4. Beyond the generative results, authors provides the empirical study about the linear performance of the proposed D-JEPA with the pixel/latent-level inputs.

**Weaknesses:**

1. The organization and the structure of the current version should be improved. The writing of chapter 3 is very confusing for me. And some important results and discussions should be re-located in the main paper not the appendix.

2. The novelity is limited:
 It seems that this work just replace the MAGE parts of MAR. The simple combination of I-JEPA and MAR's dfiffusion parts. It doesn't solve the core issue between the gap of representation learning and generative modeling.
Such an archecture cannot bring huge improvement on representation learning with latent-level input (Results in AppendixE over  down-stream tasks are good, just for pixel-level inputs), and the corresponding performance are very poor.
Could you please provide the similar settings for MAR? Use both the pixel-level and latent-level inputs to check the representation performance? I think your design with JEPA cannot solve the inherent limitation (latent input -> bad representation results) of such an TOKEN learning + diffusion framework.

3. The core of I-JEPA is the small-block-wise masks to perform the feature-level augmentation to achieve the better linear performance without the help of data-level augmentation. But in your work, the random mask strategy with high ratio is applied. Could you please provide some empirical evidence and intuitions about such a design？


4. Line 200： “It is important to note that training in latent space is not a necessity for D-JEPA; it can be trained in raw space and still achieve excellent results.” WHERE is the corresponding generative results？ I haven't found it.

5. It is unfair to compare Huge scale models in Table1 and 2. The parameters of the proposed D-JEPA is not aligned with MAR-H.

6. How about the D-JEPA-H linear and finetuning performance in Table 7/8.

7. Please provide the exps on Imagenet-512x512 to compare with  SOTAs in a more comoutation situation.

**Questions:**

Listed in Weakness.

---

> ### Author Response · Authors · 2024-11-18
> **Replying to Q1**
>
> ## The organization and the structure of the current version should be improved. The writing of chapter 3 is very confusing for me. And some important results and discussions should be re-located in the main paper not the appendix.
>
> Thank you for your suggestions on improving the structure of the paper. We will make the necessary adjustments in the revision version. Here, I would like to provide some additional clarification regarding Chapter 3: Since our work is highly related to MAR and JEPA, we provided a brief introduction to these works in Chapter 2 (Background). I believe this background information will help readers better understand the content of Chapter 3. As mentioned in the introduction, the core focus of this paper is to demonstrate the importance of representation learning in generative tasks, rather than showcasing that D-JEPA can achieve the best results in both representation learning and generative tasks. Therefore, we emphasize D-JEPA's performance in image generation in the experiments section, while its performance in representation learning is presented in the appendix.

---

> ### Author Response · Authors · 2024-11-18
> **Replying to Q2**
>
> ## The novelity is limited: It seems that this work just replace the MAGE parts of MAR. The simple combination of I-JEPA and MAR's dfiffusion parts. It doesn't solve the core issue between the gap of representation learning and generative modeling. Such an archecture cannot bring huge improvement on representation learning with latent-level input (Results in AppendixE over down-stream tasks are good, just for pixel-level inputs), and the corresponding performance are very poor. Could you please provide the similar settings for MAR? Use both the pixel-level and latent-level inputs to check the representation performance? I think your design with JEPA cannot solve the inherent limitation (latent input -> bad representation results) of such an TOKEN learning + diffusion framework.
>
>
> Firstly, D-JEPA is not aimed at unifying representation learning and generative tasks, but rather at serving as an effective framework to unify various continuous modality generation tasks, such as image, audio, and video generation. Therefore, we showcase D-JEPA's generative performance across different modalities in Appendix G. The experiments on D-JEPA's representation tasks in the appendix are supplementary, demonstrating that D-JEPA can also perform reasonably well in representation tasks. Its performance in representation learning still requires further exploration, which is beyond the scope and contribution of this paper.
>
> Secondly, regarding the comparison between MAGE and D-JEPA, MAGE does not mention any concepts related to JEPA and is based on contrastive loss for feature learning, whereas D-JEPA uses L1 loss. These are fundamentally different approaches, and it is not rigorous to confuse them due to architectural similarities.
>
> Lastly, concerning the performance of D-JEPA and MAGE in representation learning, especially in the latent space, we provided a detailed analysis in Appendix E. We attribute the suboptimal performance primarily to the pretrained VAE model's unsuitability for handling augmented image data. To support this hypothesis, we provided experimental results in the pixel space. The results (Tab 7 and Tab 8) indicate that D-JEPA trained on raw pixels can be comparable to mainstream representation learning methods.
>
> To reiterate, the core of D-JEPA is not to unify representation and generative models; achieving optimal results in representation tasks is not within the research scope of this work.

---

> ### Author Response · Authors · 2024-11-18
> **Replying to Q3**
>
> ## The core of I-JEPA is the small-block-wise masks to perform the feature-level augmentation to achieve the better linear performance without the help of data-level augmentation. But in your work, the random mask strategy with high ratio is applied. Could you please provide some empirical evidence and intuitions about such a design？
>
> Firstly, I-JEPA does not use small-block-wise masks. The masked region is an irregular polygon, covering up to 80% of the area. In D-JEPA, we randomly mask tokens as the smallest unit, with a mask ratio between 0.75 and 1.0. Thus, there is no significant difference in the masked area between the two. Regarding the choice of mask ratio in D-JEPA, we mainly followed the configurations of MAGE/MAR/MAE. Additionally, during experiments, we found that maintaining a relatively high mask ratio significantly reduces computational load and improves convergence speed. Therefore, we adhered to the configurations of previous works.

---

> ### Author Response · Authors · 2024-11-18
> **Replying to Q4**
>
> ## Line 200： “It is important to note that training in latent space is not a necessity for D-JEPA; it can be trained in raw space and still achieve excellent results.” WHERE is the corresponding generative results？ I haven't found it.
>
> Since our method description and experimental results throughout the paper are based on latent space, the experimental results in Appendix E are conducted in raw pixel space. The note in L200 is primarily to prevent reader confusion, ensuring they do not mistakenly believe that D-JEPA can only be trained in latent space. As for generative results in raw space, we did not evaluate them but instead demonstrated their representation task performance in Appendix E.

---

> > ### Comment · Reviewer_dQp1 · 2024-11-18
> >
> > Could you please list the generative performance of pixel+JEPA over various scale?

---

> > > ### Author Response · Authors · 2024-11-18
> > >
> > > Are you referring to the FID results of D-JEPA trained directly on raw pixels? (Because pixel+JEPA, that is, I-JEPA, is itself trained in pixel space, and it is not used for generation.)
> > >
> > > Currently, we are unable to provide immediate FID results for D-JEPA of various sizes trained on raw pixel data, primarily due to two reasons:
> > >
> > > 1. D-JEPA is primarily validated for its performance in latent space, which is the prevailing approach, as seen in models like MAR. Our experiments in raw space were aimed at testing the hypothesis regarding the decline in representational capability due to the VAE space, rather than achieving state-of-the-art results on raw pixels. Therefore, during the experiments described in Appendix E, we conducted only limited training for all D-JEPA models in raw space, making it difficult to provide precise FID results at this time.
> > >
> > > 2. We have not conducted experiments on D-JEPA-H in raw space, mainly for two reasons. Firstly, mainstream representation learning methods are primarily based on ViT-B/L. Secondly, the experimental cost of training a D-JEPA-H model in raw space is prohibitively high.
> > >
> > > Nonetheless, we plan to conduct comprehensive training for D-JEPA-B and D-JEPA-L models on raw pixels and subsequently update the FID results in this comment thread. We appreciate your patience and encourage you to keep an eye out for these updates.

---

> > > > ### Author Response · Authors · 2024-11-23
> > > >
> > > > Thank you for your interest in our work and for your patience.
> > > >
> > > > We have completed the experiments in raw space, and in the revision version, we have included the generative results of D-JEPA in raw space in Appendix E.2. In summary, both D-JEPA-B and D-JEPA-L achieved results in raw space that are similar to those obtained with semantic tokens. Due to the significant computational demands, we did not conduct experiments for D-JEPA-H in raw space.

---

> ### Author Response · Authors · 2024-11-18
> **Replying to Q5**
>
> ## It is unfair to compare Huge scale models in Table1 and 2. The parameters of the proposed D-JEPA is not aligned with MAR-H.
>
> For a fair comparison between D-JEPA-H and MAR-H, please refer to the detailed discussion [here](https://openreview.net/forum?id=d4njmzM7jf&noteId=Vh2BYSKA09). In short, the D-JEPA-L model, with only 687M parameters and 480 epochs of training, already surpassed MAR-H (943M parameters) with 800 epochs of training (FID 2.32 vs. 2.35). This sufficiently demonstrates that D-JEPA outperforms MAR.

---

> ### Author Response · Authors · 2024-11-18
> **Replying to Q6**
>
> ## How about the D-JEPA-H linear and finetuning performance in Table 7/8.
>
> We did not conduct experiments on D-JEPA-H for representation learning because D-JEPA-H is based on ViT-H. Current representation learning tasks typically do not use ViT-H as a backbone, mainly opting for ViT-B and ViT-L. Therefore, to ensure a fair comparison, we presented the results for D-JEPA-B and D-JEPA-L.

---

> ### Author Response · Authors · 2024-11-18
> **Replying to Q7**
>
> ## Please provide the exps on Imagenet-512x512 to compare with SOTAs in a more comoutation situation.
>
> To better showcase the effectiveness of D-JEPA, we provide the experimental results of D-JEPA-L on ImageNet-512:
>
> |   Model   | FID(w/o cfg) | FID(w/ cfg) |
> |:---------:|:------------:|:-----------:|
> |  MaskGIT  |     7.32     |             |
> | MAGVIT-v2 |     3.07     |    1.91     |
> | DiT-XL/2  |    12.03     |    3.04     |
> |   GIVT    |     8.35     |             |
> | EDM2-XXL  |     1.91     |    1.81     |
> |   MAR-L   |     2.74     |    1.73     |
> | D-JEPA-L  |     1.89     |    1.62     |
>
> Additionally, we have showcased the text-to-image generation results of D-JEPA at 1024 resolution [here](https://openreview.net/forum?id=d4njmzM7jf&noteId=houkfsPAB9).

---

> ### Author Response · Authors · 2024-11-25
> **Heartfelt Thanks and Revised Manuscript Submission**
>
> Dear Reviewer dQp1,
>
> I hope this message finds you well. On behalf of our research team, I would like to express our sincere gratitude for the time and effort you have dedicated to reviewing our manuscript. Your insightful comments and constructive feedback have been invaluable in guiding our revisions and improving the clarity and quality of our work.
>
> We have carefully addressed each of your comments and have made detailed revisions to the manuscript, particularly in the introduction and experimental analysis sections, to ensure clarity and comprehensiveness. We hope that these changes effectively address your concerns and enhance the overall presentation of our research.
>
> We kindly invite you to review the revised version of our manuscript. If there are any remaining concerns or additional feedback, please do not hesitate to reach out. We are eager to engage in further discussions and make any necessary adjustments to meet the high standards expected by the committee.
>
> Thank you once again for your invaluable input and for considering the revisions we have made. We hope that our efforts will encourage you to reassess the manuscript favorably.
>
> Warm regards

---

> > ### Comment · Reviewer_dQp1 · 2024-11-25
> >
> > Thank the author for the rebuttals.
> > My main concerns have been addressed, so I increase my score to 6.
> > Looking forward for you open-sourced codebase and models.

---

> ### Comment · Reviewer_dQp1 · 2024-11-25
>
> After detailed consideration, i am leaning to give the score of 5.
> Reasons:
> 1. The newly added ablation studies (Table 6) present that there is no obvious improvements after adding the JEPA loss.
>     And more importantly, neither CFG or IS results are provided.
>
> 2. Even though the 512x512 results of D-JEPA in Table8 are good, the parameters of D-JEPA-L is much more than MAR-L. And the detailed training epochs are not provided (I guess D-JEPA 480 and MAR 800).
>
> 3. I would appreciate the fair comparisons with MAR series, with same parameters and same epochs, not current 1/3 more parameters and 1/2 less training epochs. Such an comparison is unfair.
>
> The total representation results in my rating of 5.

---

> > ### Author Response · Authors · 2024-11-25
> > **Addressing Concerns and Updates on Our Revised Manuscript**
> >
> > We sincerely appreciate your continued interest in our work and are grateful for your insightful feedback.
> >
> > 1. **Ablation Studies and Metrics:**
> >
> >    We provided the FID metric without CFG because we believe it is the most representative and widely focused upon metric. During training, we primarily use this metric to evaluate model performance. The IS metric serves as an auxiliary reference, and we rarely rely on it to determine model quality. Regarding the absence of CFG-related FID and IS metrics, we explained in the second paragraph of Appendix D.1 (around line 1285) that different models have varying sensitivities to CFG, making FID and IS without CFG potentially more convincing. Based on these considerations, we initially provided only the FID metric without CFG in the ablation studies.
> >
> >    We truly appreciate your concern, which prompted us to reassess the relevant checkpoints and include both IS metrics and metrics without CFG in Table 6. Below is the updated Table 6, which has been revised in the latest version:
> >
> >    | Model    | \#params | Epoch | FID(w/o cfg) | IS(w/o cfg) | FID(w/ cfg) | IS(w/ cfg) |
> >    |----------|----------|:-----:|:------------:|:-----------:|:-----------:|:----------:|
> >    | MAR-B    | 208M     |  800  |     3.48     |    192.4    |     2.31    |    281.7   |
> >    | D-JEPA-B | 208M+4M  |  600  |     3.50     |    189.2    |     2.25    |    279.8   |
> >    |          |          |  800  |     3.43     |    195.3    |     2.01    |    280.1   |
> >    |          |          |  1400 |     3.40     |    197.1    |     1.87    |    282.5   |
> >
> >    As shown, D-JEPA-B demonstrates significant advantages over MAR-B. The FID (w/o cfg) metric converges well at 600 epochs, with subsequent improvements mainly in the FID (w/ cfg) metric, where we show significant advantages. The FID (w/o cfg) metric sees no significant improvement beyond 600 epochs due to model convergence.
> >
> > 2. **Training Epochs and Parameter Comparisons:**
> >
> >    In line 1341, we clarified that all experimental settings follow the latent space configuration, meaning D-JEPA was trained for 480 epochs (this will be noted below Table 8's title in the revised version for easier comparison). Regarding the parameter discrepancy between D-JEPA-L and MAR-L, we previously explained [here](https://openreview.net/forum?id=d4njmzM7jf&noteId=B3X0OHeuS9) that our use of the standard ViT-L structure results in parameter differences. However, in the 256x256 experiments, D-JEPA-L (687M) surpasses all models, including MAR-H (943M), in both convergence speed and final performance, demonstrating D-JEPA's superiority over MAR.
> >
> >    Additionally, under the FID (w/o cfg) metric, only D-JEPA-L achieves a score below 2 (1.89), with the only other method under 2 being EDM2-XXL (w/o cfg FID=1.91) with 1.5B parameters, further highlighting D-JEPA-L's significant advantage.
> >
> >    We understand your concern about potential performance bottlenecks at higher resolutions. Therefore, in our latest [supplementary material](https://openreview.net/attachment?id=d4njmzM7jf&name=supplementary_material), we provide 1k images generated by D-JEPA for T2I tasks and images of arbitrary continuous resolutions. We also detail the comparison between D-JEPA and Fluid in the *About the Revision and Supplementary Material* section under *Compared with Fluid*. Fluid is an extension of MAR for T2I tasks by the MAR authors. Our supplementary samples clearly show that D-JEPA's 2.6B model significantly outperforms Fluid's 10.5B model, serving as a reference to alleviate your concerns.
> >
> > 3. **Fair Comparisons with MAR Series:**
> >
> >    We apologize for any confusion caused by the current experimental setup. Please understand our constraints: training models at 512 resolution requires substantial computational resources. Due to time and other limitations, we cannot rerun the experiments. Furthermore, D-JEPA's parameter configuration follows the ViT series to ensure fair comparison in subsequent representation learning experiments. MAR authors may have different considerations, as they do not need to focus on representation learning experiments. Forcing alignment of parameter sizes between two different methods seems unreasonable.
> >
> > We sincerely thank you for your continued interest in our work and for your valuable suggestions for improvement. We have updated the relevant content in the paper and hope the above responses address your concerns. Additionally, we hope you recognize that D-JEPA's experiments on ImageNet are just one application scenario for this architecture. Its potential in speech, video, and multimodal understanding and generation, as shown in Appendix G, makes D-JEPA worthy of widespread attention. According to community feedback, at least MAR cannot be successfully applied to speech generation, as detailed [here](https://github.com/LTH14/mar/issues/35).

---

> ### Author Response · Authors · 2024-11-25
>
> Dear Reviewer dQp1,
>
> I hope this email finds you in good spirits. We are grateful for your detailed evaluation and have worked hard to address your concerns. With the discussion deadline fast approaching, we would appreciate it if you could revisit the manuscript to see if the adjustments align with your expectations. Your expertise continues to play a pivotal role in refining our work.
>
> Thank you once again for your dedication.
>
> Sincerely,
>
> Authors

---

### Official Review · Reviewer_gTwo · 2024-11-03

**Soundness:** 3
**Presentation:** 3
**Contribution:** 3
**Rating:** 6
**Confidence:** 3

**Summary:**

This paper extends the JEPA framework—a masked-image-based self-supervised representation learning model—to image generation by incorporating an additional MLP head that operates on image patches. This design unifies representation learning and image generation within a single framework, demonstrating improved image generation quality over recent state-of-the-art methods in specific model configurations.

**Strengths:**

(1) This paper introduces a straightforward and effective approach to enhance generation quality by bridging representation learning and image generation.
(2) In the appendix, the authors offer an in-depth analysis of the model design, including additional insights on representation learning as well as applications in video and audio generation, showcasing the versatility of the proposed methods across multiple tasks.
(3) The presentation is clear, and the ideas are easy to understand.

**Weaknesses:**

(1) The performance is comparable to similar approaches like MAR, which does not use the JEPA loss. In Table 1, the proposed method requires more training epochs (1400 vs. 800 for D-JEPA-B VS ) to achieve similar results to MAR-B. While D-JEPA-L/H outperforms MAR-L/H, it also involves more parameters. Similar trends are observed in Table 2.
(2) There is no comparison to baseline methods, such as the effect of removing the JEPA loss.
(3) How does the model perform in unconditional generation tasks or on more complex datasets, such as COCO?

**Questions:**

My main concerns with this paper relate to its performance compared to the similar approach, MAR. The absence of ablation studies on model components and evaluation on other datasets makes it challenging to assess the impact of each design choice.

---

> ### Author Response · Authors · 2024-11-18
> **Replying to Q1**
>
> ## The performance is comparable to similar approaches like MAR, which does not use the JEPA loss. In Table 1, the proposed method requires more training epochs (1400 vs. 800 for D-JEPA-B VS ) to achieve similar results to MAR-B. While D-JEPA-L/H outperforms MAR-L/H, it also involves more parameters. Similar trends are observed in Table 2.
>
>
> 1. Actually, our performance far exceeds MAR. For a detailed comparison and analysis regarding the parameter count and fair comparison between D-JEPA-L/H and MAR-L/H, please refer to the detailed discussion [here](https://openreview.net/forum?id=d4njmzM7jf&noteId=Vh2BYSKA09).
>
> 2. Regarding the experiments of D-JEPA-B in Tab.1, we indeed ran for 1400 epochs, which is 600 more than MAR-B. However, the D-JEPA-B model already achieved comparable results to MAR-B at 600 epochs (FID 3.50 vs. 3.48), as shown in Fig. 6. The reason we continued to 1400 epochs was to further explore the performance ceiling of D-JEPA-B through sufficient training. The final experiments also demonstrate that while D-JEPA-B converges well at 600 epochs, its performance continues to improve with additional training.
>
> 3. Unlike D-JEPA, MAR is currently unable to achieve effective results in speech generation tasks, as analyzed [here](https://github.com/LTH14/mar/issues/35). However, the D-JEPA architecture can handle tasks such as speech and video generation. We have showcased generated speech segments on our homepage: https://d-jepa.github.io/.

---

> ### Author Response · Authors · 2024-11-18
> **Replying to Q2**
>
> ## There is no comparison to baseline methods, such as the effect of removing the JEPA loss.
>
> There is indeed an ablation study in the paper, although not explicitly compared and analyzed. Here we provide the corresponding analysis. We conducted ablation experiments on the D-JEPA-B model. The results of MAR-B serve as the baseline (i.e., the results without the JEPA Loss), because MAR-B and D-JEPA-B have identical model structures and experimental parameters, making them a comparable baseline.
>
> From Fig. 6, we can see that D-JEPA-B achieves the performance of MAR-B at 600 epochs. Additionally, from Tab.1 and Tab.2, we observe that D-JEPA-B significantly outperforms MAR-B at 1400 epochs. For a more intuitive comparison, we provide some key data points below:
>
> |  Model   | Epoch |  FID  |
> | :------: | :---: | :---: |
> |  MAR-B   |  800  | 3.48  |
> | D-JEPA-B |  600  | 3.50  |
> | D-JEPA-B |  800  | 3.43  |
> | D-JEPA-B | 1400  | 3.40  |
>
> These results are obtained without using classifier-free guidance, which we believe better reflects the true scenario.

---

> ### Author Response · Authors · 2024-11-18
> **Replying to Q3**
>
> ## How does the model perform in unconditional generation tasks or on more complex datasets, such as COCO?
>
> For performance on larger and more complex datasets, please refer to the analysis provided [here](https://openreview.net/forum?id=d4njmzM7jf&noteId=WMuPFBJ94j) and [here](https://openreview.net/forum?id=d4njmzM7jf&noteId=houkfsPAB9). The D-JEPA model not only excels in complex text-to-image tasks but also performs well in multimodal tasks.

---

> ### Author Response · Authors · 2024-11-18
> **Replying to Concerns**
>
> ## My main concerns with this paper relate to its performance compared to the similar approach, MAR. The absence of ablation studies on model components and evaluation on other datasets makes it challenging to assess the impact of each design choice.
>
> I hope the above responses address your concerns. Overall, while D-JEPA is similar to MAR, its core lies in utilizing a representation learning architecture combined with a next-token prediction mechanism. These aspects significantly enhance D-JEPA's performance. Moreover, the design of D-JEPA significantly surpasses the performance of REPA. REPA is a recent work that also aims to integrate representation learning, but its overall performance is much lower than D-JEPA, as analyzed [here](https://openreview.net/forum?id=d4njmzM7jf&noteId=29qgdHLajO).

---

> ### Author Response · Authors · 2024-11-25
> **Heartfelt Thanks and Revised Manuscript Submission**
>
> Dear Reviewer gTwo,
>
> I hope this message finds you well. On behalf of our research team, I would like to express our sincere gratitude for the time and effort you have dedicated to reviewing our manuscript. Your insightful comments and constructive feedback have been invaluable in guiding our revisions and improving the clarity and quality of our work.
>
> We have carefully addressed each of your comments and have made detailed revisions to the manuscript, particularly in the introduction and experimental analysis sections, to ensure clarity and comprehensiveness. We hope that these changes effectively address your concerns and enhance the overall presentation of our research.
>
> We kindly invite you to review the revised version of our manuscript. If there are any remaining concerns or additional feedback, please do not hesitate to reach out. We are eager to engage in further discussions and make any necessary adjustments to meet the high standards expected by the committee.
>
> Thank you once again for your invaluable input and for considering the revisions we have made. We hope that our efforts will encourage you to reassess the manuscript favorably.
>
> Warm regards

---

> > ### Author Response · Authors · 2024-11-25
> >
> > Dear Reviewer gTwo:
> >
> > Thanks a lot for your efforts in reviewing this paper. We tried our best to address the abovementioned concerns. As the discussion deadline between reviewers and authors is very close, we tend to confirm whether there are unclear explanations and descriptions here. We could further clarify them.
> >
> > Thanks!
> >
> > Authors

---

> ### Author Response · Authors · 2024-11-25
>
> Dear Reviewer gTwo,
>
> I hope this email finds you in good spirits. We are grateful for your detailed evaluation and have worked hard to address your concerns. With the discussion deadline fast approaching, we would appreciate it if you could revisit the manuscript to see if the adjustments align with your expectations. Your expertise continues to play a pivotal role in refining our work.
>
> Thank you once again for your dedication.
>
> Sincerely,
>
> Authors

---

> > ### Comment · Reviewer_gTwo · 2024-11-26
> > **Thank authors for the feedback**
> >
> > I appreciate the authors' detailed explanations and providing additional experiments. The new experiments comparing the approach to MAR have largely addressed my concerns, and I am increasing my score to 6.

---

### Author Response · Authors · 2024-11-13
**Preface**

# Preface

We would like to extend our heartfelt thanks to each reviewer for their valuable comments. Before we proceed with a detailed, point-by-point response, I hope that the reviewers and the Area Chair could spare a moment to browse through this blog content. It primarily addresses potential misunderstandings regarding our paper and provides comprehensive comparisons with recent methodologies (those published on ArXiv after the ICLR submission deadline).

---

> ### Author Response · Authors · 2024-11-13
> **Regarding Model Scale and Experimental Results**
>
> ## Regarding Model Scale and Experimental Results
>
> Both Reviewer gTwo and Reviewer dQp1 have expressed concerns about the fairness of the experimental comparisons in Tab.1. They noted that D-JEPA-H has more parameters compared to MAR-H (1.4B vs. 943M). Consequently, although D-JEPA-H achieved superior results with just 320 epochs of training compared to MAR-H’s 800 epochs (FID 2.04 vs. 2.35), they question the fairness of this comparison.
>
> Firstly, let us clarify the principle behind the D-JEPA model configurations at different scales. To fairly complete the representation learning experiments in Appendix E, we chose to base our work on the existing ViT-B/L/H frameworks. As shown in Table 7, existing tasks focusing solely on representation learning have been based on the standard ViT B/L/H. Thus, our D-JEPA B/L/H models respectively adopt the corresponding ViT B/L/H as the backbone, resulting in parameter sizes of 212M, 687M, and 1.4B. For the MAR series models, MAR-B also uses the ViT-B structure, but MAR-L/H do not follow the standard ViT-L/H, which restricts our ability to maintain parameter consistency.
>
> Secondly, concerning the experiments in Table 1, it should be noted that our D-JEPA-L model already outperforms all previous works (including MAR-H) with 480 epochs of training and 687M parameters. This adequately illustrates the superiority of the D-JEPA architecture, even without considering the results of D-JEPA-H. Nevertheless, we trained the D-JEPA-H model to further demonstrate the scaling laws of the D-JEPA architecture, and unsurprisingly, it achieved even better results. Detailed analysis on this part is provided in lines 375 to 409 of the paper. Below is an excerpt from Tab 1, offering a more intuitive comparison:
>
> |          | #Params | #Epochs |  FID  |  IS   | Pre.  | Rec.  |
> | :------: | :-----: | :-----: | :---: | :---: | :---: | :---: |
> |  MAR-L   |  479M   |   800   | 2.60  | 221.4 | 0.79  | 0.60  |
> | D-JEPA-L |  687M   |   480   | 2.32  | 233.5 | 0.79  | 0.62  |
> |  MAR-H   |  943M   |   800   | 2.35  | 227.8 | 0.79  | 0.62  |
> | D-JEPA-H |  1.4B   |   320   | 2.04  | 239.3 | 0.79  | 0.62  |

---

> ### Author Response · Authors · 2024-11-13
> **Comprehensive Comparisons with Recent Methodologies**
>
> ## Comprehensive Comparisons with Recent Methodologies
>
> We have noticed that several highly relevant works have been published on ArXiv after the ICLR submission deadline. According to the guidelines, we are not required to compare with these works, but to fully demonstrate the superiority of the D-JEPA architecture, we have conducted a comprehensive comparison here.
>
> The core idea of D-JEPA is to bring the advantages of representation learning into generative tasks, rather than pursuing a unified model for both representation learning and generative tasks, which Reviewer dQp1 has misunderstood. Based on this premise, REPA has made similar discoveries. REPA effectively shortened the training time of DiT by adding additional feature learning tasks. D-JEPA, on the other hand, is a novel architecture that, apart from being used for image generation, has more crucial potential roles. Firstly, it is capable of generating across various continuous modalities (something both MAR and REPA cannot achieve), including audio and video. Secondly, D-JEPA employs a Next-token-prediction approach, laying the groundwork for constructing unified multimodal large models in the future. These two key differences are at the heart of the D-JEPA work. For a better comparison with REPA, we present the results in the table below:
>
> |                 | #Params | #Iters |  FID  |
> | :-------------: | :-----: | :----: | :---: |
> | SiT-XL/2 + REPA |  675M   |  400K  |  7.9  |
> | SiT-XL/2 + REPA |  675M   |   1M   |  6.4  |
> | SiT-XL/2 + REPA |  675M   |   4M   |  5.9  |
> |    D-JEPA-L     |  687M   |  300K  | 2.32  |
>
> As seen in the table above, D-JEPA-L, with a comparable parameter size, achieves results far superior to REPA with 4M iterations using only 300K iterations. This clearly demonstrates the superiority of the D-JEPA-L architecture.
>
> REPA: Representation Alignment for Generation: Training Diffusion Transformers Is Easier Than You Think, [https://arxiv.org/abs/2410.06940](https://arxiv.org/abs/2410.06940)

---

> ### Author Response · Authors · 2024-11-13
> **More Experiments on T2I**
>
> ## More Experiments on T2I
>
> Recently, we have extended the T2I experiments in section G.2. We built a 2.6B text-to-image model based on D-JEPA, and on the GenEval metrics, we have surpassed the results of the highly representative diffusion model, SD3.0 (4B version) (0.65 vs. 0.64). The detailed metrics are listed in the table below:
>
> |        | #Params | Overall | Single Obj | Two Obj | Counting | Colors | Position | Color Attr |
> |:------:|:-------:|:-------:|:----------:|:-------:|:--------:|:------:|:--------:|:----------:|
> | SD3.0  |  2.0B   |  0.62   |    0.98    |  0.74   |   0.63   |  0.67  |   0.34   |    0.36    |
> | D-JEPA |  2.6B   |  0.65   |    0.98    |  0.80   |   0.59   |  0.87  |   0.22   |    0.47    |
> | SD3.0  |  4.0B   |  0.64   |    0.96    |  0.80   |   0.65   |  0.73  |   0.33   |    0.37    |
>
> It is noteworthy that this is the first work to surpass diffusion models in the T2I task using a next-token prediction-based architecture. This achievement lays a solid foundation for the future design of unified multimodal model architectures.

---

> > ### Author Response · Authors · 2024-11-18
> > **More examples on T2I**
> >
> > To fully demonstrate the potential of D-JEPA, we present some high-resolution images generated on T2I here, to show that D-JEPA is not only limited to generating images with a resolution of 256, but also capable of generating very good high-resolution images.
> >
> > (Please click [here](https://anonymous.4open.science/r/DJEPA-T2I-Examples-389C/README.md) if the link below is invalid.)
> >
> > Prompt: [A young girl‘s face disintegrates while beautiful colors fill her features, depicted in fluid dynamic brushwork with colorful dream-like illustrations.](https://anonymous.4open.science/r/DJEPA-T2I-Examples-389C/examples/001.jpg)
> >
> > Prompt: [An Asian girl, black hair, white T-shirt, fresh style, urban background.](https://anonymous.4open.science/r/DJEPA-T2I-Examples-389C/examples/002.jpg)
> >
> > Prompt: [Art collection style and fashion shoot, in the style of made of glass, dark blue and light pink, paul rand, solarpunk, camille vivier, beth didonato hair, barbiecore, hyper-realistic.](https://anonymous.4open.science/r/DJEPA-T2I-Examples-389C/examples/003.jpg)
> >
> > Prompt: [Three spheres made of glass falling into ocean. Water is splashing. Sun is setting.](https://anonymous.4open.science/r/DJEPA-T2I-Examples-389C/examples/004.jpg)

---

> ### Author Response · Authors · 2024-11-13
> **Conclusion**
>
> ## Conclusion
>
> We sincerely appreciate the valuable comments from the reviewers. After the paper concludes the double-blind review process, we will open-source all the code and models for D-JEPA, including the code for audio, video, and multimodal generation as mentioned in Appendix G.

---

### Author Response · Authors · 2024-11-23
**About the Revision and Supplementary Material**

Thank you very much for continued attention to our work and for the valuable feedback provided.

Based on the feedback received during the rebuttal period, we have submitted a revised version of the paper along with additional supplementary materials. Below are some of the major revisions, and we hope that the reviewers and the Area Chair will take a fresh look at the latest version:

1. We have added comparisons with VAR (Visual Autoregressive Modeling: Scalable Image Generation via Next-Scale Prediction) in Table 2.
2. Following the suggestions from Reviewer 5MfA, we have revised the inappropriate expressions in the Related Work section of Appendix A. We have also made careful revisions to the introduction and experiment sections of the paper, emphasizing the performance of generative modeling.
3. In response to Reviewer dQp1's comments, we have included more experimental data in Appendices C.3, D.2, and E.2, including generative results at 512 resolution, results in raw space, and ablation studies on the JEPA Loss.

**Compared with Fluid** Additionally, during the rebuttal phase, we noticed that several reviewers were particularly interested in comparisons between D-JEPA and MAR. Recently, we have observed new work based on MAR, namely Fluid (SCALING AUTOREGRESSIVE TEXT-TO-IMAGE GENERATIVE MODELS WITH CONTINUOUS TOKENS), which extends MAR to a model size of 10.5B for text-to-image tasks. To better address concerns about D-JEPA's performance in more complex tasks, we have provided comparison results in the latest supplementary materials between images generated by the D-JEPA (2.6B) model and the Fluid (10.5B) model in text-to-image tasks. It is evident that the image quality generated by D-JEPA is significantly higher than that of Fluid.

**More results** Furthermore, to address concerns about D-JEPA's image generation capabilities beyond 256 resolution, we have included more high-resolution (1k) generation results and results across arbitrary resolutions (ranging from 64 to 1024) in the latest supplementary materials. As these results extend beyond the scope of the current work, we have not included them in the paper's appendix but are providing them as [supplementary materials](https://openreview.net/attachment?id=d4njmzM7jf&name=supplementary_material) for the reviewers' consideration.

---

### Author Response · Authors · 2024-11-26
**Rebuttal Summary**

Dear Area Chair and Reviewers,

We would like to express our heartfelt gratitude for the time and effort you have dedicated to reviewing our manuscript. We are especially thankful to reviewers dQp1, 3tNA, and 5MfA for their active participation throughout the rebuttal phase. We sincerely hope that reviewers dQp1 and gTwo will re-evaluate the contents of our revised version, and if there are still any concerns, we encourage you to provide feedback promptly. We are committed to addressing any issues as quickly as possible. Your insightful comments and constructive feedback have been invaluable in guiding our revisions and improving the clarity and quality of our work.

**Key Revisions and Responses:**

1. **Ablation Studies and Additional Results:**
   - We have included more detailed results and analyses of the ablation studies in Appendix C.3 and Table 6.
   - Additional experiments and analyses on ImageNet 512x512 have been added to Appendix D.2 and Table 8.
   - Experiments on D-JEPA's generative modeling in raw pixel space are now included in Appendix E.2 and Table 11.

2. **Other Modifications to the Paper:**
   - We have made more precise revisions to the main content of the paper to eliminate potential misunderstandings.
   - Appropriate modifications have also been made to the Related Work section in Appendix A.

**Broader Impact and Potential:**

Beyond ImageNet experiments, D-JEPA demonstrates significant potential in speech, video, and multimodal understanding and generation tasks, as showcased in Appendix G. The model's versatility and effectiveness in these domains highlight its broader impact and the value it brings to the field.

Considering the reviewers' concerns about D-JEPA's performance on more complex tasks, we have provided additional experimental results and comprehensive comparisons in the [supplementary materials](https://openreview.net/attachment?id=d4njmzM7jf&name=supplementary_material) and [Preface](https://openreview.net/forum?id=d4njmzM7jf&noteId=B3X0OHeuS9) sections. These include comparisons with recent works such as Fluid-10.5B, a continuation of MAR, and REPA, which have recently been posted on ArXiv.

We hope that our detailed responses and the revisions made to the manuscript address your concerns and demonstrate the robustness and significance of our work.

Thank you once again for your invaluable feedback and support.

Warm regards,

The Authors

---

### Meta-Review · Area_Chair_rhcE · 2024-12-18

**Metareview:**

The paper introduces a new architecture called Joint-Embedding Predictive Architecture (JEPA), which aims to enhance generative modeling tasks. The authors have demonstrated that JEPA achieves state-of-the-art performance on the ImageNet benchmarks. Moreover, its potential extends to various domains such as speech, video, and multimodal tasks, highlighting its versatility and scalability.

The paper's primary strength lies in its innovative approach of incorporating JEPA into generative modeling, marking a significant departure from traditional methods. The extensive experimentation and the results further support the model's capabilities in different domains.

The paper has faced criticism regarding the perceived incremental nature of integrating Masked Autoencoders (MAE) and Masked Autoregressive (MAR) methods. Additionally, there was some initial confusion about the model's approach to representation learning and generation. Concerns were also raised about the fairness of experimental comparisons due to discrepancies in training specifics and parameter sizes.

The authors have effectively addressed the concerns about the integration of MAE and MAR by providing additional results and explanations. The confusion regarding the model's unified approach has been cleared up with a revised focus on generative capabilities. Furthermore, the authors have discussed the parameter size differences in detail and provided supporting results, which helps to validate the model's performance.

**Additional Comments On Reviewer Discussion:**

During the review process, the main points raised included the incremental benefit and novelty of the MAE and MAR integration (Reviewer 5MfA), the clarity of the model's unified approach (Reviewer 5MfA), and the fairness in experimental comparisons due to parameter size differences (Reviewer dQp1).

The authors' rebuttal included additional results and clarifications for the MAE and MAR integration, which resolved the concerns from Reviewer 5MfA. They also revised the manuscript to clarify the primary focus on generative capabilities, which addressed the confusion pointed out by Reviewer 5MfA. Concerns raised by Reviewer dQp1 regarding parameter size differences and training specifics were addressed in detail with additional supporting results.

In reaching the final decision, I have weighed the concerns against the authors' responses. The concerns raised by Reviewer 5MfA have been satisfactorily resolved through the rebuttal. As for the parameter size differences, the authors have provided a compelling explanation and additional results that support the effectiveness of their model.

---

### Decision · Program_Chairs · 2025-01-22

Accept (Poster)